# END-TO-END LEARNING UNDER ENDOGENOUS UNCERTAINTY

## ABSTRACT

How can we effectively learn to make decisions when there are no ground-truth counterfactual observations? We propose an end-to-end learning approach to the contextual stochastic optimization problem under decision-dependent uncertainty. We propose both exact methods and efficient sampling-based methods to implement our approach. We also introduce a new class of two-stage stochastic optimization problems to the end-to-end learning framework. Here, the first stage is an information-gathering problem to decide which random variable to "poll" and gain information about before making a second-stage decision based off of it. We provide theoretical analysis showing (1) that optimally minimizing our proposed objective produces optimal decisions and (2) generalization bounds between in-sample and out-of-sample cost. We computationally test the proposed approach on multi-item assortment problems where demand is affected by cross-item complementary and supplementary effects. Overall, our method outperforms other benchmarks by more than 15% and performs best in high noise, across any cost configuration, and when given sufficient data. We also introduce an experiment for the information-gathering problem on a real-world electricity generation problem. We show our method proposes decisions with more than 7% lower cost than other decision-making methods.

## 1 INTRODUCTION

We consider the general problem of contextual stochastic optimization under decision-dependent uncertainty. Often in decision-making one is faced with a two-step problem: first to predict some unknown quantity/random variable such as product demand, and second to make an operational decision based off of this such as allocating inventory. We will consider settings in which this operational decision-making process is well-defined through traditional optimization-based methods as is common in many applications ranging from pricing, to inventory allocation, to scheduling. In this paper we consider that the random variable is dependent on the decision made. For example, the demand (random variable) of a product will change depending on the price (the decision) set. We refer to this as *endogenous* uncertainty.

Here, the predictive model needs to take the decision as an input as well as additional features. In order to make a decision, one would need to optimize over the predictive model's input. This brings several challenges. (1) The more complex the learning model, the more difficult to optimize. Linear-like models are most tractable, but provide less predictive power. On the other hand, neural networks or random forests with more predictive power are significantly more expensive to optimize over. (2) When optimizing over the entire input space, it becomes easy to choose decisions that are far out-of-sample and for which the model has poor predictive power. This may result in decisions with significantly worse objective than predicted. As an example for pricing, the actual demand could be significantly lower than predicted. This is especially problematic when there is sparse or limited data. Moreover, it is often unclear what aspect of the distribution to predict. For instance, while one is interested in maximizing mean reward, it is generally not optimal to choose the decision which maximizes the reward of the mean counterfactual outcome. We will see this explicitly in the following sections as well as the experimental section. We introduce an approach to jointly predict and optimize in this endogenous setting which learns a prediction aligned with expected cost.

Existing end-to-end frameworks (all under exogenous random variables) are not able to tackle the single-stage endogenous problem The traditional end-to-end formulation requires knowledge of the outcome or value of a decision taken. When the uncertainty is independent of the decision taken, one can simply use the data as the ground truth. For example, demand is independent of supply in a warehouse. Therefore, for any decision taken based off of a learned demand prediction, one can then calculate the cost using the realized demand observed in the data. This cost is then used as the loss for the demand prediction problem. However, when the uncertainty depends on the decision, one can no longer take this approach since one does not have access to counterfactual information. That is, we do not have knowledge about what would have happened if a different action were taken.

This endogenous end-to-end problem is also closely connected to information-gathering problems which have not been studied under the end-to-end learning framework to date. In this class of problems, there is an initial stage before the prediction and decision-making step where one is allowed to gather information about some of the random variables ahead of time. For example, one can send out a survey or set up a poll to better understand demand at a particular location. Given this new information, one can then make more informed predictions about the rest of the random variables (for e.g. demand at the remaining locations), and subsequently make a more informed decision. If demand across locations is correlated, one can gain significant information from polling a single location. As a first stage, one must decide which location to poll, observe information about this location, then make a new prediction and a decision for all locations conditioned on this new observation. So, there are three questions (1) which random variable should we poll (2) what predictions to make conditioned on observing this chosen variable and (3) what decision to make based off of these predictions.

In contrast to the first endogenous uncertainty setting we presented, the random variables here are not dependent on the decisions we take. However, our knowledge of the random variable does depend on the first stage polling decision we must make. We extend our proposed method for end-to-end learning with endogenous random variables to this two-stage information-gathering setting. In this paper, we propose a new method of applying the end-to-end ideology to this setting of endogenous uncertainty and the closely related information-gathering problem. We present the following contributions:

1. We formulate an end-to-end, or joint prediction and optimization, approach when the realizations of uncertainty are dependent on the decisions taken. The objective is to train a model whose corresponding predictions on in-sample decisions have task-based reward close to the observed reward in-sample. See section 2.
2. We provide theoretical analysis showing that (1) optimally minimizing our proposed objective produces optimal decisions and (2) convergence bounds on the generalization gap with respect to the amount of training data used and model complexity. Overall, the generalization gap between in-sample and out-of-sample cost decreases as $1/\sqrt{N}$ where $N$ is the amount of training data.
3. Due to the non-convex nature of the end-to-end objective, we provide mixed-integer optimization formulations, as well as a computationally efficient sampling-based approach. See Appendix A.
4. We extend our proposed method to the information-gathering problem. This is a combination of both traditional end-to-end methods under exogenous variables in the second-stage problem and under endogenous variables in the first stage problem. This two-stage problem class has not been studied under the end-to-end learning setting to date.
5. Finally, we show computational experiments on a multi-item assortment optimization problem where the demand of a product is dependent on decisions taken for all other items due to complementary or supplementary effects. We show the end-to-end approach improves significantly on traditional two-stage methods. We also consider an electricity generation and scheduling problem. Here we make an initial forecast, and must decide on the optimal time to update the forecast. This involves learning how to balance the benefits of waiting for more accurate information against the costs of delaying decisions. See section 4.

**Related Work.** Within the space of end-to-end offline contextual stochastic optimization there has been relatively little work in the case of endogenous uncertainty for general optimization problems. For instance, Bertsimas & Koduri (2022) primarily focuses on exogenous uncertainty and briefly mentions endogenous uncertainty as an extension by adding the historical decision as an additional feature to use to learn the outcome. However, this does not take into account the pitfalls we mentioned earlier. Endogenous uncertainty has been studied primarily for specific problems such as pricing Liu

& Zhang (2023) where demand naturally changes according to price, or a facility location problem Basciftci et al. (2021) where demand changes depending on where a facility is placed. However these take significantly different approaches from ours, not learning any parameteric model to predict uncertainty, or ignoring learning goals from that of decision making (an aspect we refer to as two-stage, or predict-then-optimize). Within the scope of exogenous uncertainty, where the decision does not affect the uncertainty, there is a variety of work addressing different classes of objective functions and single or two-stage/multi-stage problems. This includes the work of Elmachtoub & Grigas (2022), Amos & Kolter (2017), Agrawal et al. (2019) and more. We refer the reader to surveys Kotary et al. (2021); Sadana et al. (2023) for a more comprehensive survey.

Online learning and multi-armed bandit problems also have a similar problem setting. For example, contextual linear bandits Chu et al. (2011) or more recently the estimation-to-decision-making meta-algorithm Foster & Rakhlin (2020) focus on learning the decision-dependent reward of actions. There are a variety of extensions relevant to our work, like continuous action spaces, Majzoubi et al. (2020), Krishnamurthy et al. (2020) or large action space Dulac-Arnold et al. (2015) in reinforcement learning. Even more similar to our scenario is offline learning or offline reinforcement learning Lange et al. (2012), Levine et al. (2020). In the end-to-end setting we consider, we are given more information about the problem structure than in usual bandit/RL problems. Specifically, there is some intermediate random variable that one observes (such as demand, while reward corresponds to revenue) for a given action (such as inventory allocation). Our approach makes explicit use of this additional structure and this is one of the main reasons we observe better performance than methods that directly learn reward. Work on performative learning, such as Perdomo et al. (2020), focuses on how predictions themselves may affect observations (such as how traffic predictions will affect driver behavior and hence traffic itself). This work focuses on how the distribution shifts over time, depending on how predictions are made.

In contrast to more common reward-learning approaches, we consider the offline, not the online learning setting. Learning the relationship between decisions and the intermediate random variable also has additional advantages. Specifically, we may have additional domain knowledge about this relationship. For example, demand of a product is generally a monotonically decreasing function of price. This additional structure can be imposed on the learning process. However, learning revenue directly as a function of price does not exhibit such structure we can take advantage of. We give a more in-depth comparison with other methods in the next section.

The problem we consider is essentially a contextual stochastic programming problem with decision-dependent uncertainty. In the non-contextual case, there has been significant work on developing methods to solve these complex problems. See for example Goel & Grossmann (2006), Dupacová (2006). These are difficult-to-solve problems, even in the no-context case when explicitly knowing the distribution of the random variable. The main advantage of an end-to-end approach is in reducing this complexity. Instead of making a distributional prediction of the relationship between decision and the random variable, we make a deterministic one. One can view our end-to-end approach as learning which deterministic prediction will lead to the same decision compared to when making a distributional one. We prove in proposition 2.1 such a prediction does exist under mild assumptions.

Versions of the information-gathering problem have been studied as well. The most closely related line of work is in the area of value of information by Howard (1966). This aims to decide the amount a decision maker would be willing to pay for information prior to making a decision. This notion of value information is particularly relevant in reinforcement learning applications, deciding which actions to explore to gain the most useful information (for e.g. Arumugam & Van Roy 2021). This work still differs significantly as ours also considers the specific structure of the objective and intermediate random variables as we explained previously.

## 2 PROBLEM SETTING AND RELATED END-TO-END LEARNING LITERATURE

We first formally describe the problem. One wishes to makes decisions $\mathbf{w} \in \mathcal{P}$ in feasible region $\mathcal{P} \subset \mathbb{R}^d$. Associated with this decision is an objective $g(\mathbf{w}, \mathbf{z})$ which is a function of a random variable $\mathbf{z} \in \mathbb{R}^p$ dependent on the decision $\mathbf{w}$ itself and additional contextual information $\mathbf{x}$. We say $\mathbf{z}$ is distributed according to some unknown distribution $Z(\mathbf{w}, \mathbf{x})$. We give two example. (1) $\mathbf{w}$ is the price chosen for a product and $z$ is the uncertain demand that depends on the price. Then, $g(\mathbf{w}, \mathbf{z}) = \mathbf{w} \cdot \mathbf{z}$. And (2) *Assortment*: Uncertain demand $\mathbf{z}$ depends on the shelf-space inventory $\mathbf{w}$

displayed. Now consider the objective to minimize $g(\mathbf{w}, \mathbf{z}) = \max\{\mathbf{z} - \mathbf{w}, 0\} + c \cdot \mathbf{w}$ for paying a backorder cost for each unit of unfulfilled demand, and a unit cost of $c$ for procuring each unit.

We are given offline data consisting of information $(\mathbf{x}^1, \mathbf{w}^1, \mathbf{z}^1), \dots, (\mathbf{x}^N, \mathbf{w}^N, \mathbf{z}^N)$ of features $\mathbf{x}_n$, decisions $\mathbf{w}_n$ (which are potentially suboptimal) and observed uncertainty $\mathbf{z}^n \sim Z(\mathbf{w}^n, \mathbf{x}^n)$. Finally, our objective is to learn some function $f(\mathbf{w}, \mathbf{x})$ that predicts some statistic of the uncertainty/random variable $\mathbf{z}$ in such a way that the corresponding decisions have maximum objective. Given $f$, and some out-of-sample feature $\mathbf{x}$, one takes decision given by finding $\mathbf{w}$ that maximizes the objective:

$$\hat{w}(\mathbf{x}) = \arg\max_{\mathbf{w} \in \mathcal{P}} g(\mathbf{w}, f(\mathbf{w}, \mathbf{x})). \tag{1}$$

One is interested in learning some relationship between decisions $\mathbf{w}$, features $\mathbf{x}$, and the random variable $\mathbf{z}$. Ideally, we would have access to some function $f^*(\mathbf{w}, \mathbf{x})$ for which

$$g(\mathbf{w}, f^*(\mathbf{w}, \mathbf{x})) = \mathbb{E}_{\mathbf{z} \sim Z(\mathbf{w}, \mathbf{x})}[g(\mathbf{w}, \mathbf{z})]. \tag{2}$$

Then, solving equation 1 using $f^*$ would exactly find the optimal solution. A-priori, it is unclear whether such an $f^*$ exists in the first place. We show $f^*$ indeed exists in proposition 2.1. The proof can be found in Appendix B.

**Proposition 2.1.** *For continuous objective functions $g(\mathbf{w}, \mathbf{z})$ with respect to $\mathbf{z}$, there exists $\hat{\mathbf{z}}$ in the convex hull of the support of $Z(\mathbf{w}, \mathbf{x})$ so that*

$$g(\mathbf{w}, \hat{\mathbf{z}}) = \mathbb{E}_{\mathbf{z} \in Z(\mathbf{w}, \mathbf{x})}[g(\mathbf{w}, \mathbf{z})]. \tag{3}$$

We propose in this paper to learn a model which predicts some statistical function of the uncertainty ($\mathbf{z}$) so that the resulting objective value/reward of the *historical* decisions best matches the observed *historical* reward. That is, we observe $\mathbf{z}^n$ under decision $\mathbf{w}^n$ and features $\mathbf{x}_n$, with reward $g(\mathbf{w}_n, \mathbf{z}_n)$. Our goal is to match this with predicted reward $g(\mathbf{w}^n, f(\mathbf{w}^n, \mathbf{x}^n))$. Therefore, our objective is

$$\hat{f}_{\text{end-to-end}} = \arg\min_{f \in \mathcal{F}} \sum_{n=1}^{N} (g(\mathbf{w}^n, f(\mathbf{w}^n, \mathbf{x}^n)) - g(\mathbf{w}^n, \mathbf{z}^n)))^2 \tag{4}$$

where $\mathcal{F}$ is the hypothesis class of functions $f$ we learn from. Finally, once $\hat{f}_{\text{end-to-end}}$ is learned, we can solve problem equation 1 to make decisions. Problem equation 4 can be solved directly by gradient descent. We provide exact methods of solving it by mixed-integer programs and other approximation methods that may be of interest in Appendix A, although we use traditional gradient descent for simplicity in the experiments.

**Convergence.** For $\hat{f}_{\text{end-to-end}}$ to converge to the true $f^*$ as the amount of data grows, we need the hypothesis class itself to be rich enough to contain $f^*$. In proposition 2.1 we showed that an $f^*$ does exist in the first place. We can increase the complexity of $\mathcal{F}$ as needed to achieve better results. However, given limited data, increasing the complexity of $\mathcal{F}$ can also worsen out-of-sample accuracy. We bound this in the following theorem. We will first formally define the complexity of the hypothesis by Rademacher complexity.

**Definition 2.2** (Multidimensional Rademacher Complexity). The empirical Rademacher complexity of the hypothesis class of functions $\mathcal{F} : (\mathbf{w}, \mathbf{x}) \to \mathbb{R}^d$ is given by $\mathcal{R}_N(\mathcal{F}) = \mathbb{E}_{\{(\mathbf{w}^n, \mathbf{x}^n)\}_{n=1}^N} \mathbb{E}_\sigma \left[ \sup_{f \in \mathcal{F}} \frac{1}{N} \sum_{n=1}^{N} \sum_{k=1}^{d} \sigma_{nk} f_k(\mathbf{w}^n, \mathbf{x}^n) \right]$ where $\sigma_{nk}$ are i.i.d. variables uniformly sampled from $\{-1, 1\}$ (also known as Rademacher variables).

Finally, we show the following bounds.

**Theorem 2.3.** *For any function $f \in \mathcal{F}$, we define out-of-sample error/loss $l$ and the empirical loss $\hat{l}$ over a random sample of $N$ datapoints $(\mathbf{w}^1, \mathbf{x}^n, \mathbf{z}^n), n = 1, \dots, N$*

$$l(f) = \mathbb{E}_{\mathbf{w}, \mathbf{z}, \mathbf{x}}[(g(\mathbf{w}, f(\mathbf{w}, \mathbf{x})) - g(\mathbf{w}, \mathbf{z}))^2], \quad \hat{l}(f) = \sum_{n=1}^{N} (g(\mathbf{w}^n, f(\mathbf{w}^n, \mathbf{x}^n)) - g(\mathbf{w}^n, \mathbf{z}^n)))^2.$$

*For any $L$-Lipschitz function $g$ (with respect to $\mathbf{z}$), $g(\mathbf{w}, \mathbf{z}) \in [0, 1] \forall \mathbf{w} \in \mathcal{P}$ and all $\mathbf{z}$ which we assume has bounded support of $Z(\mathbf{w}, \mathbf{x})$, then we have with probability $1 - \delta$,*

$$l(f) \leq \hat{l}(f) + 2L\sqrt{2}\mathcal{R}_N(\mathcal{F}) + \left( \frac{8 \log 2/\delta}{N} \right)^2. \tag{5}$$

For many hypothesis classes $\mathcal{F}$, we can bound $\mathcal{R}_N(\mathcal{F})$ by a term that converges to 0 as $N \to \infty$ and at a rate $O(1/\sqrt{N})$ for common function classes like linear functions. See for example Bartlett & Mendelson (2002). So, the overall generalization gap decreases at a $O(1/\sqrt{N})$ rate.

In the experiments in section 4, we will benchmark against reward-learning methods (methods that only predict final reward instead of intermediate random variable $\mathbf{z}$). This is a more complex mapping to learn and we see find that indeed these reward-learning methods underperform. Next, we will compare this methodology against the traditional exogenous case in section 2.1. Here, the random variable $\mathbf{z}$ is independent of decisions $\mathbf{w}$ and is only affected by features $\mathbf{x}$. As in the exogenous end-to-end setting, the goal of $\hat{f}$ is to remove the need to compute an expectation, and approximate it with a point forecast. Once $\hat{f}_{\text{end-to-end}}$ is learned, we solve (1) to make decisions.

## 2.1 COMPARISON WITH OTHER APPROACHES

**Learning the mean.**  We remark that it is crucial to learn an $\hat{f}$ with this task-based loss. The common approach in ML would be to learn a model which learns the mean of the distribution of $z$. That is, minimize the mean-squared error between predictions and historical observations. We will denote this as a two-stage approach, first predicting $z$ independent of the task loss $g$, then optimizing for the optimal decision. The two-stage predictor, $\hat{f}_{\text{2-stage}}$, is a predictor of the mean $\mathbb{E}[\mathbf{z}]$. The issue arises in the second stage when optimizing. In general $\mathbb{E}[g(\mathbf{w}, \mathbf{z})] \neq g(\mathbf{w}, \mathbb{E}[\mathbf{z}])$ when $g$ is non-linear in $\mathbf{z}$. Therefore, optimizing $\max_{\mathbf{w} \in \mathcal{P}} g(\mathbf{w}, \hat{f}_{\text{2-stage}}(\mathbf{w}, \mathbf{x}))$ would be a proxy for optimizing $\max_{\mathbf{w} \in \mathcal{P}} g(\mathbf{w}, \mathbb{E}_{\mathbf{z} \sim Z(\mathbf{w}, \mathbf{x})}[\mathbf{z}])$ but not the true objective which is $\max_{\mathbf{w} \in \mathcal{P}} \mathbb{E}_{\mathbf{z} \sim Z(\mathbf{w}, \mathbf{x})} g(\mathbf{w}, \mathbf{z})$.

**Learning reward directly.**  Many methods in online learning such as contextual bandits or reinforcement learning learn the reward function directly, instead of the intermediate random variable $z$. This removes the issues in the previous section about learning the mean of $z$ since here we directly optimize the reward. However, there are several computational downsides to reward-learning. Here, we learn a mapping $r(\mathbf{w}, \mathbf{x}) \approx \mathbb{E}_{\mathbf{z} \sim Z(\mathbf{w}, \mathbf{x})}[g(\mathbf{w}, \mathbf{z})]$ while only observing $\mathbf{w}^n, \mathbf{x}^n$ and $g(\mathbf{w}^n, \mathbf{z}^n)$ and not $\mathbf{z}^n$ itself or the structure of the function $g$. Directly learning the reward function requires a more complex class of predictors to capture this relationship compared to an end-to-end method. Simplifying the predictor class $r$ is crucial as the complexity of $r$ directly impacts the difficulty of solving $\max_{\mathbf{w} \in \mathcal{P}} r(\mathbf{w}, \mathbf{x})$ in large/continuous action space. Our approach allows for simpler model classes while still capturing the same complexity in modeling $\mathbb{E}_{z \sim Z(\mathbf{w}, \mathbf{x})}[g(\mathbf{w}, \mathbf{z})]$. We further see this explicitly in the numerical computations, see section 4.

**Distinction from exogenous case**  Finally, we describe and compare against the setting under exogenous random variables. We will also use this methodology in conjunction with ours in the two stage information-gathering setting which we will introduce in section 3. The first stage is endogenous, while the second is exogenous. For ease of comparison, we denote all parameters in the exogenous case with a bar. Here one observes features $\mathbf{x}$ and corresponding realizations of uncertainty $\bar{\mathbf{z}}$ coming from a distribution $\bar{Z}(\mathbf{x})$ that depends only on $\mathbf{x}$. On the other hand, in the endogenous case, $\mathbf{z}$ depends on both $\mathbf{x}$ and $\mathbf{w}$. The objective in the exogenous case is

$$\max_{\mathbf{w} \in \mathcal{P}} \mathbb{E}_{\bar{\mathbf{z}} \sim \bar{Z}(\mathbf{x})}[g(\mathbf{w}, \bar{\mathbf{z}})]. \tag{6}$$

Computing this expectation is difficult. So, one goal of an end-to-end approach is to replace this with a point forecast which produces the same decision and objective value. The objective is to learn a point forecast $\bar{f}(\mathbf{x})$ to replace the distribution $\bar{Z}(\mathbf{x})$. Replacing $\bar{Z}(x)$ with the deterministic $\bar{f}(\mathbf{x})$ in problem equation 6 give us $\max_{\mathbf{w} \in \mathcal{P}} g(\mathbf{w}, \bar{f}(\mathbf{x}))$ which is computationally much simpler to solve. Given a point forecast $\mathbf{z} = \bar{f}(\mathbf{x})$ denote the optimal corresponding decision by $w^*(\mathbf{z})$:

$$w^*(\bar{\mathbf{z}}) = \arg \max_{\mathbf{w} \in \mathcal{P}} g(\mathbf{w}, \bar{\mathbf{z}}). \tag{7}$$

We would like to learn an $\bar{f}$ for which

$$\mathbb{E}_{\bar{z} \sim \bar{Z}(\mathbf{x})} \left[ g(w^*(\bar{f}(\mathbf{x})), \bar{\mathbf{z}}) \right] \approx \max_{\mathbf{w} \in \mathcal{P}} \mathbb{E}_{\bar{\mathbf{z}} \sim \bar{Z}(\mathbf{x})}[g(\mathbf{w}, \bar{\mathbf{z}})] \tag{8}$$

which allows us to replace $\bar{Z}(x)$ with $\bar{f}(x)$ in equation 6. This is similar to our proposed objective equation 2. Under exogenous variables, the common data-driven objective is to learn a model $\bar{f}$

which maximizes the reward/objective of the corresponding decisions $w^*(\bar{f}(\mathbf{x}))$ that it takes. See for example Elmachtoub & Grigas (2022). Given data $(\mathbf{x}^n, \bar{\mathbf{z}}^n)_{n=1}^N$, we wish to solve

$$\hat{f}_{\text{exo}} = \arg\max_f \sum_{n=1}^N g(w^*(\bar{f}(\mathbf{x}^n)), \bar{\mathbf{z}}^n). \tag{9}$$

After learning some $\hat{f}_{\text{exo}}$ (for the exogenous case) one then takes decisions $w^*(\hat{f}_{\text{exo}}(\mathbf{x}))$. In the endogenous setting, we predict a point statistic of the uncertainty so that the predicted reward for any action taken (including historical ones) are close to their realization. In contrast, here in the exogenous case, we are predicting a statistic of the uncertainty so that the value of the optimal decision given a point forecast is close to the optimal expected reward. The methodology used for the exogenous case cannot be applied to the endogenous case because it would require one to have access to counterfactual information. The objective value of the decision $w^*(\hat{f}_{\text{exo}}(\mathbf{x}))$ cannot be evaluated because it was not taken historically, and we hence we do not have any information on the random variable $\mathbf{z}$ that would depend on the decision. In the exogenous case, $\mathbf{z}$ does not depend on the decision, so we can indeed evaluate the decision cost by using historically observed $\mathbf{z}$.

## 3 APPLICATION TO INFORMATION GATHERING

Here we consider a novel application of the end-to-end method to a class of 2-stage optimization problems with information-gathering. As an example, consider a multi-warehouse inventory allocation problem. In the first stage, we can choose a single location to poll to learn the demand for the next time period. In the second stage, we must (1) predict the demand at all other locations, conditioned on our previous observation from the poll then (2) decide how much inventory to allocate at all warehouses. We note that this is different from traditional 2- or multi-stage stochastic optimization problems. There, the first stage is some operational decision (for e.g. inventory allocation in a warehouse), and then in the second stage some additional information is revealed (such as demand). In our setting, there is a deliberate initial action to decide which additional information to reveal ahead of time (for e.g. before allocating inventory, we can poll one location to know exact demand).

One approach would be to learn a model that predicts, for every location, the expected cost that results from polling it. This simplifies the problem but hides the structure behind it and makes the learning problem more complex, requiring a richer class of functions to approximate it. We explicitly observe the advantage of using the problem structure while learning an end-to-end model in the experiments.

Formally, we are given exogenous random variables $\mathbf{z} = (z_1, \ldots, z_d)$, independent of the decisions that we take. In the first stage, the decision space will consist of choosing some index $w \in \mathcal{P} = \{1, \ldots, d\}$ to survey, or gain more information about, the $w^{th}$ entry of $\mathbf{z}$, namely $z_w$. This could be more general beyond observing a single value, for example observing multiple values. We will see this in an experiment in section 4. But we will keep this simple here for the sake of notation. As another example, this could correspond to setting up a survey to learn more about the demand of the $w^{th}$ product. In the second stage, we make a prediction for the remaining random variables, conditioned on the observation of $z_w$. Note this prediction does depend on the decision we initially took to survey the $w^{th}$ random variable. In the second stage, we are given some auxiliary decision variables $\mathbf{v} \in \mathcal{V}$ with objective function $g(\mathbf{v}, \mathbf{z})$. In short, the entire process is as follows:

1) For an out-of-sample point $\mathbf{x}$, we make a decision $w$ to observe $z_w \sim Z_w(\mathbf{x})$.

2) Given the observation $z_w$, the full vector $\mathbf{z}$ is distributed according to $Z(\mathbf{x})|_{Z_w(\mathbf{x})=z_w}$.

3) We are now given some second-stage decision-making problem with variables $\mathbf{v} \in \mathcal{V}$ with objective $g(\mathbf{v}, \mathbf{z})$ and we wish to make decision $\mathbf{v}$ minimizing expected cost:

$$\min_{\mathbf{v} \in \mathcal{V}} \mathbb{E}_{\mathbf{z} \sim Z(\mathbf{x})|_{Z_w(\mathbf{x})=z_w}}[g(\mathbf{v}, \mathbf{z})]. \tag{10}$$

4) Ultimately, we wish to know which observational decision $w$ will minimize overall loss:

$$\min_{w \in \mathcal{P}} \mathbb{E}_{z_w \sim Z_w(\mathbf{x})} \left[ \min_{\mathbf{v} \in \mathcal{V}} \mathbb{E}_{\mathbf{z} \sim Z(\mathbf{x})|_{Z_w(\mathbf{x})=z_w}}[g(\mathbf{v}, \mathbf{z})] \right]. \tag{11}$$

In terms of data, we observe $n$ points $(\mathbf{x}^i, w^i, \mathbf{z}^i), i = 1 \ldots, n$ where $\mathbf{z}^i$ is distributed according to an (unknown) distribution $Z(\mathbf{x}^i)$. Given decision $w^i$, we observe the realization of $\mathbf{z}^i_{w^i}$ before making

---

**Algorithm 1** End-to-end information-gathering

---

Learn model $p(\mathbf{x}, z_w)$ to predict $\mathbf{z}$ conditioned on observing $z_w$.
    Learn $p$ by solving equation 14 with gradient descent.
    Compute gradients $\partial v^*(\mathbf{z})/\partial \mathbf{z}$ using any method from previous work such as Donti et al. (2017), Cristian et al. (2023), Amos & Kolter (2017).
Learn point forecast $f(w, \mathbf{x})$ by solving equation 17 by gradient descent.
For out-of-sample $\mathbf{x}$, choose decision $w$ by solving equation 18.
For out-of-sample $\mathbf{x}$ and decision $w$, observe $z_w$. And take second-stage decision $v^*(p(\mathbf{x}, z_w))$.

---

the second-stage decision. To train we proceed as follows. We first begin by simplifying the inner expectation in equation 10. After making a decision $w$ of observing $z_w$, we can make a forecast for some statistic of the distribution $Z(\mathbf{x})|_{Z_w(\mathbf{x})=z_w}$. Let $p(\mathbf{x}, z_w)$ denote this prediction for all of $\mathbf{z}$ conditioned on observing $z_w$ as well as features $\mathbf{x}$. In particular, we are now in a similar setting as the traditional end-to-end problem. For example, given forecast $p(\mathbf{x}, z_w)$ for product demand, we then need to solve an optimization problem to optimize the inventory allocation. That is, we take decision

$$v^*(p(\mathbf{x}, z_w)) = \arg \min_{\mathbf{v} \in \mathcal{V}} g(\mathbf{v}, p(\mathbf{x}, z_w)). \tag{12}$$

Essentially, we will learn these point forecasts $p(\mathbf{x}, z_w)$ in order to remove the expectations from problem equation 11. This is similar to the traditional end-to-end framework in equation 8.

i) We first learn $p(\mathbf{x}, z_w)$ to predict $\mathbf{z}$ conditioned on observing $z_w$ for the $w^{th}$ random variable. We want such a $p$ to approximate equation 10. That is, we need

$$g(v^*(p(\mathbf{x}, z_w)), \mathbf{z}) \approx \min_{\mathbf{v} \in \mathcal{V}} \mathbb{E}_{\mathbf{z} \sim Z(\mathbf{x})|_{Z_w(\mathbf{x})=z_w}} [g(\mathbf{v}, \mathbf{z})]. \tag{13}$$

We learn such a $p$ by solving the following empirical risk minimization problem, similar to equation 9:

$$\min_p \sum_{i=1}^{n} g(v^*(p(\mathbf{x}^i, z_{w^i}^i)), \mathbf{z}^i). \tag{14}$$

ii) Now, substituting equation 13 into equation 11 our final problem simplifies to

$$\min_w \mathbb{E}_{z_w \sim Z_w(\mathbf{x})} [g(v^*(p(\mathbf{x}, z_w)), \mathbf{z})]. \tag{15}$$

This problem now falls under our end-to-end framework with endogenous random variables because the objective function depends on $p(\mathbf{x}, z_w)$ which in turn depends on the first-stage decision $w$. So, similar to equation 2, we wish to learn a single point forecast $f(w, \mathbf{x})$ to replace the expectation over $\mathbf{z}$. That is, our goal is to learn a function $f$ so that

$$g\Big(v^*\big(p(\mathbf{x}, f_w(w, \mathbf{x}))\big), f(w, \mathbf{x})\Big) \approx \mathbb{E}_{z_w \sim Z_w(\mathbf{x})} [g(v^*(p(\mathbf{x}, z_w)), \mathbf{z})]. \tag{16}$$

We replace $\mathbf{z}$ with a point forecast $f(w, \mathbf{x})$. To learn $f$ we use a version of our method in equation 4:

$$\min_f \sum_{i=1}^{n} \left( g\Big(v^*\big(p(\mathbf{x}^i, f_{w^i}(w^i, \mathbf{x}^i))\big), f(w^i, \mathbf{x}^i)\Big) - g\Big(v^*\big(p(\mathbf{x}^i, z_{w^i}^i)\big), \mathbf{z}^i\Big) \right)^2. \tag{17}$$

iii) Finally, for an out-of-sample $\mathbf{x}$, we make decisions by solving

$$\arg \min_w g\Big(v^*\big(p(\mathbf{x}, f_w(w, \mathbf{x}))\big), f(w, \mathbf{x})\Big). \tag{18}$$

In practice, we cannot observe $z_w$ before making decision $w$, so in equation 17 and equation 18 we "observe" the $w^{th}$ entry of the predicted $f(w, \mathbf{x})$ instead. Algorithm 1 provides a concise description.

## 4 COMPUTATIONAL EXPERIMENTS

We present two experiments. The first is a single stage assortment problem where demand depends on inventory allocation. Second, we consider a two-stage electricity scheduling problem on real-world data. First, one makes a preliminary demand forecast, then plans to reschedule at a chosen time $t$. This $t$ must be chosen ahead of time. Afterwards, given observations up to time $t$, one makes a new forecast for the remaining time.

**Assortment Optimization** For the assortment optimization problem we are given a set of $K$ products, and we must decide the amount $w = (w_1, \ldots, w_K)$ of each product to stock. The demand of a product type depends on its own stock as well as the stock of every other item. Given a decision $w$ and demand realization $z = (z_1, \ldots, z_K)$ for each product, the cost is by $g(w, z) = \sum_{k=1}^{K} b \cdot \max\{z_k - w_k, 0\}^2 + h \cdot \max\{w_k - z_k, 0\}^2$. There is a backorder unit cost $b$ for unfulfilled demand and holding unit cost $h$ unused inventory. Further, we suppose demand of one item also depends on the presence of other items nearby. This cross-item effect is common in practice. For example, pairs of items may act as substitutes: if there is not enough of one item, customers may switch to another. Or they may act as complements: demand in one item decreases if the price of the other increases. We assume the demand of an item is a function of the stock of all other items. Each item $k$ has a base demand $\alpha_k^*$ and some perturbation based on the other items $z_k = \max\{(\alpha_k^* + \sum_{j \neq k} \beta_k^* \cdot w_k)^2 + \delta, 0\}$ where $\delta$ is gaussian noise with variance dictated by the noise level in the next experiments. The max term ensures non-negative demand. We assume a quadratic relationship between items and demand to consider a more complex learning problem.

*Methods:* We compare against the following methods (1) a *predict-then-optimize* method, also known as a *two-stage* method, which train a model to learn the uncertainty (demand in this case) as a function of actions (inventory in this case). (2) A *cost-learning* method which trains a model to learning cost directly as a function of $\mathbf{w}$. This does not take into account the intermediate demand data or the structure of the cost function. It only observes the final cost of a decision. (3) Similar to method (2), we predict cost using a *gaussian process* method instead. (4a) A K-NN-based method which, for any action, approximates cost by computing the average of the costs of the $K$-nearest neighboring actions. (4b) A K-NN method which instead predicts demand then makes decisions by solving equation 1. (5) We use our proposed method, solving equation 4. Each of the methods (1), (2), (5) use the same model architecture for making predictions. We use a feedforward neural network with a fully-connected layer of width 50. Each method is trained with the same parameters until convergence.

*Setup:* We evaluate the approaches by constructing 20 datasets by randomly iniatalizing the parameters of $\alpha, \beta$ as described earlier. We report two metrics: the average percentage error from optimality of each method on the 20 datasets (evaluated as the average of $(c - OPT)/OPT$ where $c$ is the cost of each method and $OPT$ is the optimal cost), as well as the standard deviation of these errors. We compute these metrics as vary various parameters: (i) the amount of training data available, (ii) the backorder cost to holding cost ratio, (iii) the amount of noise in the data, (iv) the width of the neural network. For brevity, we report these metrics for our approach and the next best method. Full results for all methods can be found in appendix C.1.

| | Network Depth | | | | Noise | | |
|---|---|---|---|---|---|---|---|
| Depth | Our Approach | Cost Learner | | Noise Level | Our Approach (width 50) | Cost Learner (Width 500) | K-NN (4b) |
| 10 | $1.67 \pm 0.674$ | $1.676 \pm 0.602$ | | 0.0 | $0.008 \pm 0.02$ | $0.025 \pm 0.06$ | $0.002 \pm 0.004$ |
| 30 | $2.15 \pm 0.05$ | $2.15 \pm 1.103$ | | 0.1 | $0.014 \pm 0.01$ | $0.033 \pm 0.08$ | $0.012 \pm 0.022$ |
| 100 | $0.157 \pm 0.01$ | $0.99 \pm 0.712$ | | 0.2 | $0.038 \pm 0.01$ | $0.034 \pm 0.01$ | $0.074 \pm 0.064$ |
| 500 | $0.165 \pm 0.09$ | $0.11 \pm 0.207$ | | 0.3 | $0.092 \pm 0.03$ | $0.111 \pm 0.07$ | $0.143 \pm 0.130$ |
| 1000 | $0.166 \pm 0.09$ | $0.09 \pm 0.02$ | | 0.4 | $0.157 \pm 0.04$ | $0.182 \pm 0.08$ | $0.283 \pm 0.147$ |
| | | | | 0.5 | $0.230 \pm 0.02$ | $0.366 \pm 0.23$ | $0.391 \pm 0.174$ |

Table 1: Network depth results.

Table 2: Noise results.

| | Training Data | | | | | | | | | |
|---|---|---|---|---|---|---|---|---|---|---|
| Data | 100 | 200 | 300 | 400 | 500 | 600 | 700 | 800 | 900 | 1000 |
| Our Approach | 2.83 | 1.33 | 0.04 | 0.02 | 0.03 | 0.02 | 0.02 | 0.01 | 0.02 | 0.02 |
| KNN (4b) | 0.10 | 0.10 | 0.16 | 0.17 | 0.18 | 0.13 | 0.12 | 0.13 | 0.12 | 0.13 |
| KNN (4a) | 0.44 | 0.40 | 0.34 | 0.32 | 0.31 | 0.29 | 0.21 | 0.25 | 0.17 | 0.17 |
| reward-learning | 2.36 | 2.55 | 2.31 | 1.78 | 2.11 | 1.79 | 1.90 | 1.10 | 1.15 | 0.69 |
| gaussian process | 1.76 | 2.05 | 1.78 | 1.98 | 1.41 | 1.83 | 1.70 | 1.55 | 1.87 | 1.48 |
| two-stage | 3.43 | 2.40 | 2.78 | 2.90 | 2.72 | 2.73 | 2.91 | 3.16 | 3.15 | 2.86 |

Table 3: Network depth average error (standard deviations can be found in Table 5).

In terms of parameters, we use the following setup for each set of experiments: 400 training datapoints, with 10,000 test samples to compute average cost for each decision, a backorder cost of 1, holding cost of 1, a noise level of 0.5. Each of the four experiments (i), (ii), (iii), (iv) described above will vary one of these parameters as shown in Tables 1-3 while keeping the rest of the parameters fixed.

*Results:* See Tables 1-3 for results. Full plots can be found in the appendix in section C.1. Overall, our method outperforms all other methods in high noise, across any backorder cost configuration, and when given sufficient data. We gain a few key take-aways from each of the experiments. From experiment (i) having enough data is crucial. We see a consistent improvement in our approach as data increases until it plateaus around 300 datapoints and onwards. The remaining approaches all continue to improve slowly, but even at 1000 datapoints do not reach the same level of cost as ours. See Figure 5 for a clear illustration. From experiment (ii) our method performs best across all backorder cost choices. See Table 6 in the appendix for full results. Finally from the noise experiment (iii), as we increase the noise level, our approach performs better than all other methods. This happens for several reasons. The main issue with two-stage approaches (methods (1) and (4b) described above) is the mismatch in objective — they predict the mean of the demand distribution, but this is generally the incorrect statistic to predict. For example, if we want to learn some $f(w)$ to predict demand, we wish that $g(w, f(w))$ is close to $\mathbb{E}_{z \sim D} g(w, z)$. Since $g(w, z)$ is not linear in $z$, we find that $\mathbb{E}_z g(w, z) \neq g(w, \mathbb{E}[z])$. However, when noise is low (and in particular when there is no noise at all), the objectives do match. As we increase noise, the benefit of our approach is strongly visible. The end-to-end approach learns a model $f$ which aligns $g(w, f(w))$ with $\mathbb{E}_z g(w, z)$. On the other hand, a cost-learning method (like methods (2), (3), (4a)) performs worse since one needs to learn a more complex map, whereas our approach only needs to learn a demand function. For instance, the cost learning method using a neural network with hidden layer width 500 still performs worse than our approach using a width of only 50. Full results can be found in Figure 4. Finally, (iv), we find that our approach outperforms all other methods even with a lower complexity model. For instance, using a depth of only 30, our proposed method reaches the same average cost as the cost-learning method which is only able to achieve this with depth 1000. Full results can be found in Figure 3. Standard deviations can be found in Table 5.

**Electricity scheduling: information-gathering**   We now consider the information-gathering setting introduced in section 3. We consider an electricity generation scheduling problem using data from PJM, an electricity routing company coordinating the movement of electricity throughout 13 states. The goal is to make a generation schedule and decide on the amount of electricity to generate hour per hour, over the next 24 hours. We consider the problem in two stages. First, we make an initial forecast for the 24 hours dependent only on feature information for that day. Then we decide on a time $w$ to update the schedule. Up to time $w$, we use the initial forecast to generate electricity, then given the new observations of true demand up to time $w$ we regenerate this forecast and generation schedule for the rest of the day. See figure 1 for an illustration of the sequence of events. There is now a balancing act in deciding what hour $w$ to change the schedule. If we wait longer, we gain a better estimate of future demand, however we also use a worse forecast up to the waiting time $w$. Finally, we define the objective function. The operator incurs a unit cost $\gamma_e$ for excess generation and a cost $\gamma_s$ for shortages. The cost of generating $v_1, \ldots, v_{24}$ while true demand is $z_1, \ldots, z_{24}$ is given by $g(\mathbf{v}, \mathbf{z}) = \sum_{i=1}^{24} \gamma_s \max\{z_i - v_i, 0\} + \gamma_e \max\{v_i - z_i, 0\}$.

*Methods:* Full details can be found in Appendix C.2. There are three components for each model: (1) how to make the initial forecast, (2) which time $w$ to choose, (3) how to update the forecast given $w$. We first introduce two baselines which always choose $w = 0$, so they never observe any of the day's demand, and only use their initial forecast. (1) We consider a predict-*then*-optimize approach, where we learn a demand function independent of the decision-making step. We refer to this as "Predict then optimize." (2) We learn an end-to-end model which aims to directly minimize decision cost $g$ as in equation 6. We refer to this as "Vanilla E2E" (vanilla end-to-end). We then introduce baselines that choose $w$ in different ways, including our proposed approach. Each of these methods use the same model for the initial forecast, and for making the updated forecast (components (1) and (3) above). We choose the vanilla end-to-end method for this initial forecast since it performs significantly better than the 2-stage method. We only vary the method to decide $w$. The goal is to

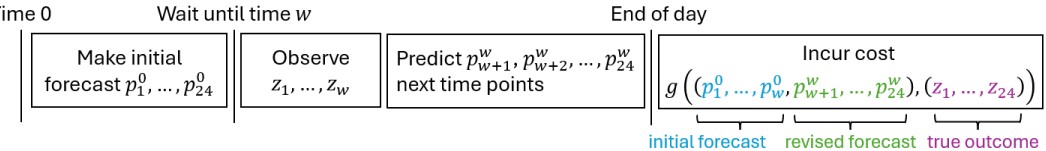

Figure 1: Electricity scheduling: sequence of actions.

| Method | Average difference | Median cost | % Endo. E2E Wins |
|---|---|---|---|
| Predict then optimize | 710% | 0.588 | 66% |
| Vanilla E2E | 55% | 0.339 | 90% |
| Cost Learner | 7.5% | 0.220 | 71% |
| **Endogenous E2E** | 0% | **0.204** | 100% |
| Random | 21% | 0.264 | 88% |
| Single action | 20% | 0.261 | 81% |
| Optimal | -7.9% | 0.187 | NA |

Table 4: Electricity scheduling: cost comparison across methods.

highlight the differences in objective cost resulting from various methods of choosing $w$. Here we train a model $p$ to predict future demand given observations $z_1, \ldots, z_w$, as well as features $\mathbf{x}$. Note that here we observe all variables up to time $w$, which is different than in section 3 where we observe a single variable. Up to time $w$, the baseline decisions are made by the vanilla end-to-end approach. After time $w$, the schedule is made according to $p$, based on *true* demand up to time $w$.

We have three baselines for methods on choosing $w$. (1) choosing a (uniformly) random action $w$. We refer to this as "Random." (2) We fix a single $w$ for all data points (choosing this best $w$ from training data). We refer to this as "Single action." Finally, (3) the optimal in-hindsight decision $w$ which may change for every out-of-sample data point $\mathbf{x}$. We refer to this as "Optimal." We will denote our proposed method to decide $w$ as "Endogenous E2E" (endogenous end-to-end). This entails solving eq. equation 17 by gradient descent for $f$ and choosing decision $w$ by solving eq. equation 18. As a final baseline, we also compare against a more traditional approach: for each decision $w$ and features $\mathbf{x}$, predict directly the loss/cost of this decision. This does not take into account the structure of the problem and simply minimizes mean-squared error between predicted cost and observed cost of each action on the training data. We refer to this as "Cost Learner." *Results:* In table 4, we report the average difference between decision cost of our method and the other methods for each datapoint. Our approach is 7.5% better than the cost-learning method and less than 8% worse than optimal on average. We also measure the median cost of each method, as well as the percentage of test datapoints on which our approach performs better than every other approach. For example, our method only outperforms the predict-then-optimize method 66% of the time, indicating this method does well on some data, but on also performs significantly more poorly on others, likely where it proposes a shortage (not knowing that a shortage is significantly worse than excess, since mean-squared error loss is unaware of this). In addition, we also plot the cost distribution of each method on the test data in Figure 2 alongside the optimal cost distribution. We observe that our proposed method most closely aligns with the optimal cost everywhere. Knowing the additional structure of the problem, our approach can better learn it.

**Conclusion**  This paper introduces an end-to-end, or joint prediction and optimization, framework for contextual stochastic optimization problems with decision-dependent uncertainty as well as for a class of two-stage information-gathering problems. This work introduces two new broad problem classes to the end-to-end framework. We evaluate our proposed method on two experiments, including one using real-world electricity demand data, and show it consistently outperforms other baselines.

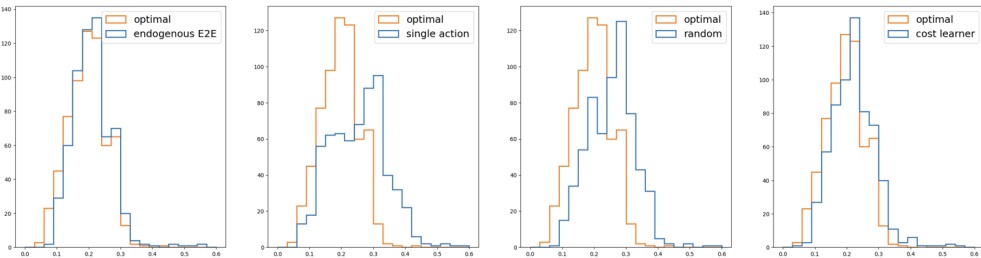

Figure 2: Cost distribution of each method. From left to right: endogenous E2E, single action, random, cost-learner.

## REPRODUCIBILITY STATEMENT

All experiments and implementations of our methods can be found in the supplementary material. This includes data, model hyperparameter choices and so on. The appendix contains significant detail on the implementation as well, both of our method and of the baselines we compare against. We clearly state whether the baselines and datasets were from pre-existing papers which have publicly available code and data. Finally, all theoretical results in the main paper have complete proofs in the appendix as well as additional algorithmic methods.

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

## A  ALGORITHMIC METHODS

### A.1  EXACT REFORMULATION

We turn to an important discussion on the difficulty and the methods to solve the learning problems presented in the earlier sections. The main difficulty in solving equation 4 is its non-convexity. Note that each term term $(g(w_n, z) - r_n)^2$ is a non-convex function of $z$ whenever $g(w, z)$ is non-linear in $z$. We first present an exact integer optimization-based formulation for solving the non-convex problem, then a more efficient sampling-based approximation. In the former exact case, we will assume that $g$ is piece-wise linear and convex, while in the latter we make no such assumption.

We first formulate equation 4 as a mixed-integer quadratic optimization problem. Let $g$ be the maximum (in the case of convex $g$) of $K$ linear functions $g^1, \ldots, g^K$ so that

$$g(w, z) = \max_{k=1,\ldots,K} g^k(w, z) \tag{19}$$

The rest of the argument also follows through if we assume that $g$ is the minimum of $K$ linear functions instead. As examples, the joint pricing and inventory allocation and assortment examples in the beginning of section 2 have this structure. Now, equation 4 can be formulated as:

$$\min_{f,v,y} \sum_{n=1}^{N} v_n \quad \text{subject to} \tag{20}$$

$$v_n \geq (g^k(w_n, f(w_n, x_n)) - g(w_n, z_n))^2 - M(1 - y_{n,k})$$

$$g^k(w_n, f(w_n, x_n)) \geq g^j(w_n, f(w_n, x_n)) - M(1 - y_{n,k})$$

$$\sum_{k=1}^{K} y_{n,k} = 1, \forall n, \quad \text{and} \quad y_{n,k} \in \{0, 1\}.$$

The binary variable $y_{n,k}$ is forced to equal 1 for any $f$ such that $g$ is equal to $g^k$. We have $g^k(w_n, f(w_n, x_n)) \geq g^j(w_n, f(w_n, x_n))$ for all $j = 1, \ldots, K$ for exactly one index $k$, hence we can set $y_{n,k} = 1$ and the constraints hold. For every other $k$, the constraints do not hold. However, since $y_{n,k} = 0$ the constraint $g^k(w_n, f(w_n, x_n)) \geq g^j(w_n, f(w_n, x_n)) - M$ do hold for large enough $M$. Finally, we force exactly one $y_{n,k}$ to equal 1 by $\sum_k y_{n,k} = 1$.

Finally, $v_n$ is simply equal to $(g^k(w_n, f(w_n, x_n)) - g(w_n, z_n))^2$ for the appropriate $k$ where $g^k = g$. Indeed, for $y_{n,k} = 1$, the first constraint becomes equivalent to $v_n \geq (g(w_n, f(w_n, x_n)) - g(w_n, z_n))^2$. Since the objective function is to minimize $\sum_n v_n$, it follows that $v_n$ will take the smallest possible value which will be equal to the maximum of all $g((w_n, f(w_n, x_n)) - g(w_n, z_n))^2 - M(1 - y_{n,k})$. Whenever $y_{n,k} = 0$, the constraints can essentially be ignored since they impose a smaller lower bound. So the maximum is achieved at $k$ for which $y_{n,k} = 1$, making $v_n = (g(w_n, f(w_n, x_n)) - g(w_n, z_n))^2$.

If $f(w, x)$ is a linear function, then the above formulation is a mixed integer quadratic-convex optimization problem and can be solved by off-the-shelf solvers. Of course, one can use augmented features and kernel functions to increase the expressivity of the prediction model while remaining linear.

### A.2 Sampling approximation

While exact, the formulation presented in the previous subsection is intractable as the amount of data increases. Here we provide two sampling-based approaches that are computationally more efficient, albeit do not guarantee optimality.

**Perturbing the mean-squared predictor:** (1) First compute the two-stage approximator (by solving $\min_\theta \sum (f_\theta(w_n, x_n) - z_n)^2$) and let $\hat{\theta}$ be the weights found. (2) For each sample $s = 1, \ldots, S$, perturb the weights $\hat{\theta}$ by some random gaussian noise $\delta^s$ to produce a sample $\theta^s = \hat{\theta} + \delta^s$. (3) Perform gradient descent using each sample $\theta^s$ as an initialization point. Choose the model with the best in-sample *task-based* loss.

**Iterative learning:** (1) First, create new smaller datasets, with the $k^{th}$ one containing the first $k \cdot D$ datapoints $(x_1, w_1, z_1), \ldots, (x_{k \cdot D}, w_{kD}, z_{kD})$ where $k = 1, \ldots, K = N/D$. (2) For $k = 1$, apply the previous method by sampling from perturbing the mean-squared predictor. Let $\theta^1$ be the final model. (3) For $k > 1$, use $\theta^{k-1}$ as the initial model, generate $S$ new samples by perturbing $\theta^{k-1}$, then apply gradient descent to minimize task-based loss. Choose $\theta^k$ with the best loss from these. Finally, (4) return the model with $\theta^D$.

The iterative learning method is essentially a super-set of the first method. We observe it provides generally better results as well in the numerical experiments. Intuitively, this makes sense: as we add more data, we fine-tune the previous model learned. Moreover, this is also useful when data arrives online. One observes data up to a time point, then make a new decision, then observe the outcome. This new observation becomes a new datapoint that can be used for training.

---

**Algorithm 2** Endogenous end-to-end

---

Learn point forecast $f(\mathbf{w}, \mathbf{x})$ by solving equation 4.
    If $g$ piece-wise linear, solve by exact method (see equation 20).
    Else, solve by sampling method in A.
For out-of-sample $\mathbf{x}$, take decisions by solving equation 1.
    If $\mathcal{P}$, small, solve by enumerating all $\mathbf{w} \in \mathcal{P}$.
    Otherwise solve by gradient descent, or traditional optimization methods.

---

# B    PROOFS

*Proof of Proposition 2.1.* Consider two values $\mathbf{z}^1$ and $\mathbf{z}^2$ so that

$$g(\mathbf{w}, \mathbf{z}^1) \leq \mathbb{E}_{\mathbf{z} \in Z(\mathbf{w}, \mathbf{x})}[g(\mathbf{w}, \mathbf{z})] \leq g(\mathbf{w}, \mathbf{z}^2). \tag{21}$$

Since $g$ is a continuous function with respect to $\mathbf{z}$, there must exist a convex combination of $\mathbf{z}^1, \mathbf{z}^2$, say $\hat{\mathbf{z}}$ so that

$$g(\mathbf{w}, \hat{\mathbf{z}}) = \mathbb{E}_{\mathbf{z} \in Z(\mathbf{w}, \mathbf{x})}[g(\mathbf{w}, \mathbf{z})]. \tag{22}$$

$\square$

*Proof of Theorem 2.3.* The results of Bartlett & Mendelson (2002) can be applied directly to the composite cost function $c(\hat{\mathbf{z}}) = (g(\mathbf{w}, \hat{\mathbf{z}}) - \mathbf{y})^2$ where for simplicity we use $\mathbf{y}$ to replace the constant $g(\mathbf{w}, \mathbf{z})$. The loss of a model $f \in \mathcal{F}$ is given by the $c \circ f = c(f(\mathbf{w}, \mathbf{x}))$. Theorem 8 of Bartlett & Mendelson (2002) gives us

$$l(f) \leq \hat{l}(f) + \mathcal{R}_N(c \circ \mathcal{F}) + \left( \frac{8 \log 2/\delta}{N} \right)^{1/2}. \tag{23}$$

Next, using the vector contraction inequality from Bartlett & Mendelson (2002), we can further bound the Rademacher complexity by

$$\mathcal{R}_N(c \circ \mathcal{F}) \leq \sqrt{2}\lambda \mathcal{R}_N(\mathcal{F}) \tag{24}$$

where the cost function $c(\hat{\mathbf{z}}) = (g(\mathbf{w}, \hat{\mathbf{z}}) - \mathbf{y})^2$ is $\lambda$-Lipschitz with respect to $\hat{\mathbf{z}}$. It remains to bound $\lambda$. Any continuously differentiable function over a compact domain is Lipschitz continuous with Lipschitz constant equal to the maximum magnitude of the derivative over that domain. In our case, $g(\cdot, \cdot) \in [0, 1]$.

We can further decompose $c(\hat{z})$ into $c_1 \circ c_2$ where $c_1(\mathbf{z}') = (\mathbf{z}')^2$ and $c_2(\hat{\mathbf{z}}) = g(\mathbf{w}, \hat{\mathbf{z}}) - \mathbf{y}$. The Lipschitz constant of $c(\hat{\mathbf{z}})$ is then bounded by the product of the Lispchitz constants of $c_1$ and $c_2$. By assumption (in theorem 2.3), $g$ is $L$-Lipschitz and hence so is $c_2(\hat{\mathbf{z}})$. Moreover, $\mathbf{z}' = c_1(\hat{\mathbf{z}}) \in [-1, 1]$ since both $g(\cdot, \cdot)$ and $\mathbf{y}$ are in $[0, 1]$. Next, $c_1$ is 2-Lipschitz since its gradient is $2\mathbf{z}'$ and its greatest magnitude is $|2\mathbf{z}'| \leq 2$ over $\mathbf{z}' \in [-1, 1]$. Therefore, $\lambda \leq 2 \cdot L$. This combined with equation 24 and equation 23 proves our theorem. $\square$

## C EXPERIMENTS

Here we give more details on the experiments as well as the exact formulations used for these problems.

### C.1 ASSORTMENT

We provide full results of the experiment results from section 4.

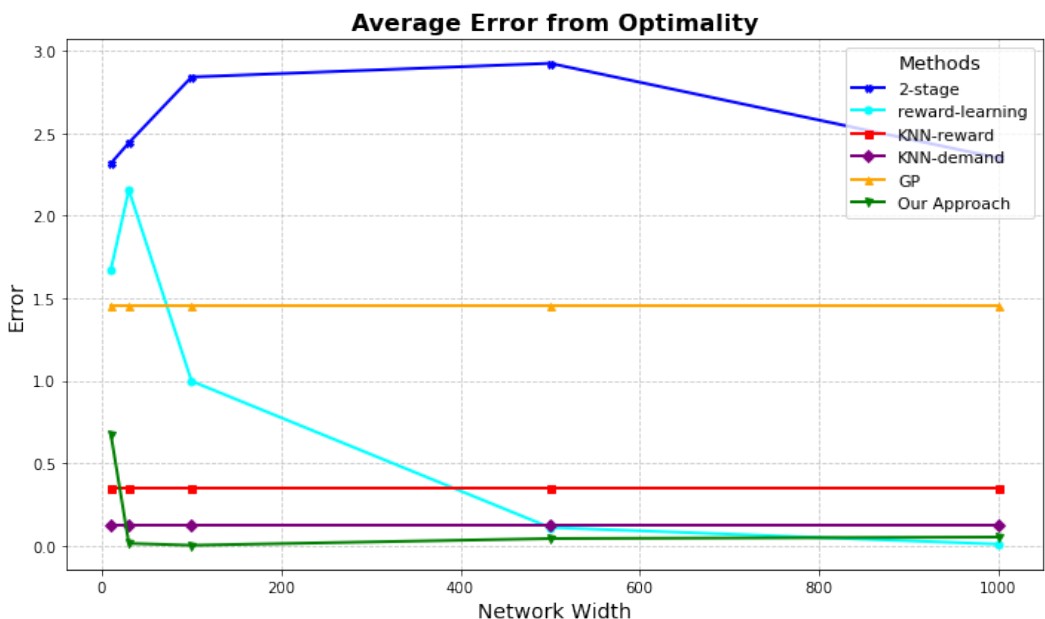

Figure 3: Average percent difference of each method from optimality as network width increases.

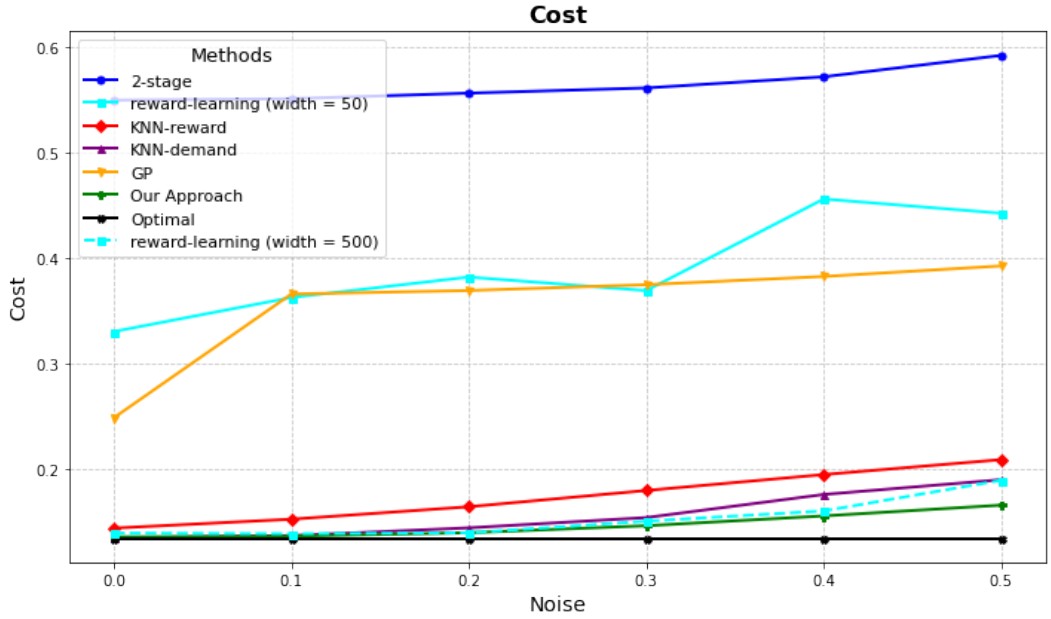

Figure 4: Average cost of each method across noise levels.

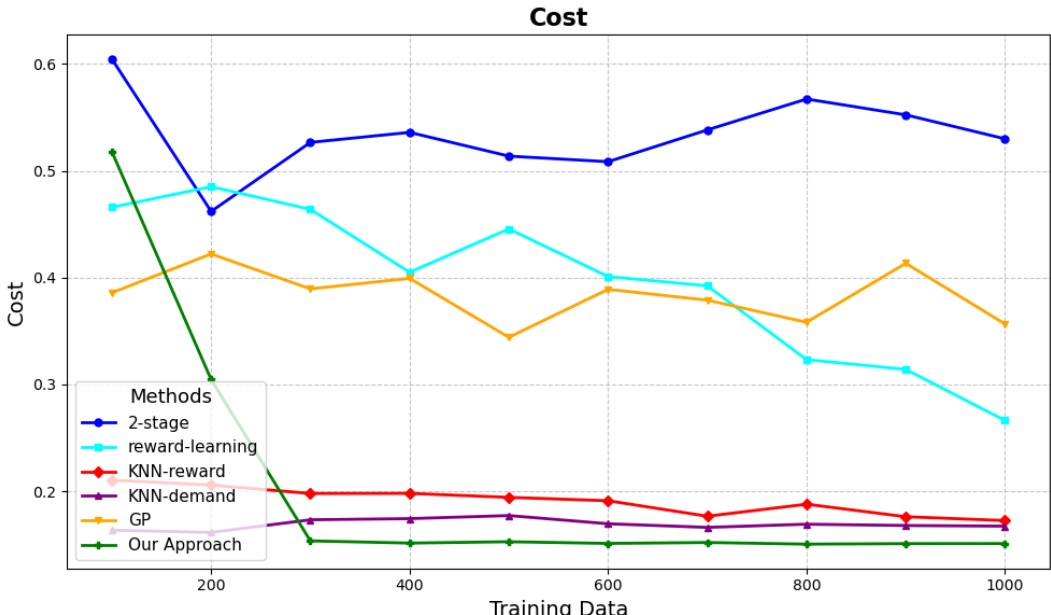

Figure 5: Results of average cost for all methods as data increases.

| Data | Two-Stage | Cost-Learning | KNN (4a) | KNN (4b) | Gaussian Process | Our Approach |
|------|-----------|---------------|----------|----------|------------------|--------------|
| 100 | $3.43 \pm 1.12$ | $2.36 \pm 0.99$ | $0.44 \pm 0.29$ | $0.10 \pm 0.12$ | $1.76 \pm 0.68$ | $2.83 \pm 1.44$ |
| 200 | $2.40 \pm 1.56$ | $2.55 \pm 0.93$ | $0.40 \pm 0.22$ | $0.10 \pm 0.13$ | $2.05 \pm 0.97$ | $1.33 \pm 1.84$ |
| 300 | $2.78 \pm 1.00$ | $2.31 \pm 0.67$ | $0.34 \pm 0.21$ | $0.16 \pm 0.17$ | $1.78 \pm 0.82$ | $0.04 \pm 0.06$ |
| 400 | $2.90 \pm 1.01$ | $1.78 \pm 0.96$ | $0.32 \pm 0.17$ | $0.17 \pm 0.16$ | $1.98 \pm 1.12$ | $0.02 \pm 0.03$ |
| 500 | $2.72 \pm 1.20$ | $2.11 \pm 0.85$ | $0.31 \pm 0.31$ | $0.18 \pm 0.23$ | $1.41 \pm 0.64$ | $0.03 \pm 0.03$ |
| 600 | $2.73 \pm 1.24$ | $1.79 \pm 0.92$ | $0.29 \pm 0.18$ | $0.13 \pm 0.12$ | $1.83 \pm 1.01$ | $0.02 \pm 0.03$ |
| 700 | $2.91 \pm 1.22$ | $1.90 \pm 1.52$ | $0.21 \pm 0.19$ | $0.12 \pm 0.09$ | $1.70 \pm 0.79$ | $0.02 \pm 0.03$ |
| 800 | $3.16 \pm 1.17$ | $1.10 \pm 0.75$ | $0.25 \pm 0.18$ | $0.13 \pm 0.09$ | $1.55 \pm 0.72$ | $0.01 \pm 0.02$ |
| 900 | $3.15 \pm 1.43$ | $1.15 \pm 0.74$ | $0.17 \pm 0.13$ | $0.12 \pm 0.09$ | $1.87 \pm 0.78$ | $0.02 \pm 0.03$ |
| 1000 | $2.86 \pm 1.25$ | $0.69 \pm 0.56$ | $0.17 \pm 0.14$ | $0.13 \pm 0.13$ | $1.48 \pm 0.78$ | $0.02 \pm 0.03$ |

Table 5: Average error and standard deviations as training data increases.

| Backorder Cost | Two-Stage | Cost-Learning | KNN (4a) | KNN (4b) | Gaussian Process | Our Approach |
|----------------|-----------|---------------|----------|----------|------------------|--------------|
| 3 | $2.56 \pm 0.86$ | $1.84 \pm 0.57$ | $0.35 \pm 0.25$ | $0.12 \pm 0.15$ | $1.45 \pm 0.62$ | $0.01 \pm 0.02$ |
| 5 | $2.71 \pm 1.14$ | $2.48 \pm 0.81$ | $0.35 \pm 0.24$ | $0.12 \pm 0.15$ | $1.45 \pm 0.62$ | $0.001 \pm 0.00$ |
| 7 | $2.70 \pm 0.83$ | $3.15 \pm 0.96$ | $0.35 \pm 0.24$ | $0.12 \pm 0.15$ | $1.45 \pm 0.62$ | $0.001 \pm 0.00$ |

Table 6: Average percentage error from optimality and standard deviations as backorder cost increases.

## C.2 INFORMATION-GATHERING: ELECTRICITY SCHEDULING

We use a similar set-up for the problem as in Donti et al. (2017). This paper only considered the pure single-stage end-to-end task, and not the two-stage problem we are considering here with information gathering. Nevertheless, we use the same data, model architecture, and similar problem parameters which we describe here.

*The set-up:* We are asked to decide/plan on the amount of electricity to generate each hour for the next 24 hours. We denote these decisions by $v_1, \ldots, v_{24}$. Given a demand realization of $z_1, \ldots, z24$,

the cost of the decision is

$$g_{\mathbf{z}}(\mathbf{v}) = \sum_{i=1}^{24} \gamma_s \max\{z_i - v_i, 0\} + \gamma_e \max\{v_i - z_i, 0\}. \tag{25}$$

Each day, we are given contextual information $\mathbf{x}^n$ and demand observation $\mathbf{d}^n$. As features, we use the past day's electrical load as well as temperature, and the temperature forecast for the current day. In addition, we use non-linear functions of the temperature, one-hot-encodings of holidays and weekends, and yearly sinusoidal features. Like the paper Donti et al. (2017), we use a 2-layer feed-forward network, both hidden layers having width 200, and an additional residual connection from the input to the output. This linear layer is initialized by first solving a linear regression problem to predict demand (this is done independently of the objective $g$).

This is the original single-stage end-to-end problem. In our setting we also consider the possibility to update the schedule as the day progresses. Each day, we must decide ahead of time a particular hour to regenerate the schedule. We denote this decision by $w \in \{0, 1, \ldots, 24\}$. For the first $w$ hours of the day, we use the original decisions made the day before. After observing $z_1, \ldots, z_w$, we then make a new forecast conditioned on these observations, and make a new decision based off of this for the remaining hours $w + 1, \ldots, 24$.

*Methods:* The base prediction model for each method is the same across all approaches. We use the two-layer network described in the above paragraphs. Let $\mathcal{F}$ denote this architecture and the set of models/weights using this architecture. Any model $f \in \mathcal{F}$ takes as input features $\mathbf{x}$ and outputs a vector in $\mathbb{R}_{\geq 0}^{24}$ for each hour of the next day.

1. **Two-Stage**: This is a simple regression model which predicts demand as a function of features $\mathbf{x}$. That is, we train

$$f_{\text{2-stage}} = \arg\min_{f \in \mathcal{F}} \sum_{n=1}^{N} (f(\mathbf{x}^n) - \mathbf{z}^n)^2. \tag{26}$$

   Then, for an out-of-sample $\mathbf{x}$, we simply set decisions $v_1, \ldots, v_n$ according to $f_{\text{2-stage}}(\mathbf{x})$.

2. **Vanilla End-to-End**: Here the objective is to directly minimize the cost of the decisions we take. So, instead of minimizing mean-squared error, the objective is to minimize cost:

$$f_{\text{end-to-end}} = \arg\min_{f \in \mathcal{F}} \sum_{n=1}^{N} g(f(\mathbf{x}^n), \mathbf{z}^n). \tag{27}$$

   Again, we take decisions decisions $v_1, \ldots, v_n$ according to $f_{\text{2-stage}}(\mathbf{x})$.

3. **Learning $p$**: Before describing the remaining methods, we first focus on the model $p(\mathbf{x}, z_1, \ldots, z_w)$. Here, we have chosen to wait until hour $w$, then we observe $z_1, \ldots, z_w$. Given these, would like to make a new forecast for the remaining hours of the day. This forecast is given by $p(\mathbf{x}, z_1, \ldots, z_w)$. Since the input length of $\hat{p}$ varies with $w$, we take the full vector $\mathbf{z}$ as input (which always has fixed length 24) and mask the time points from $w + 1$ to 24. Moreover, $\hat{p}$ still outputs the full 24 time points, but the loss function will only be evaluated on time points $w + 1$ to 24.

   We first use the initial vanilla end-to-end predictions as a baseline. Let these be $f_{\text{end-to-end}}(\mathbf{x})$. Then, the model $p$ will predict a perturbation to this baseline, dependent only on the new observations $z_1, \ldots, z_w$. Specifically, let

$$p(\mathbf{x}, z_1, \ldots, z_w) = f_{\text{end-to-end}}(\mathbf{x}) + \hat{p}(z_1, \ldots, z_w) \tag{28}$$

   and we wish to learn this $\hat{p}(z_1, \ldots, z_w)$. For our experiments, we let $\hat{p}$ be a single hidden layer network of width 200 with relu activation.

   At each batch of training, the loss function is given by

$$\frac{1}{24 - w} g_{w+1,\ldots,24}(f_{\text{end-to-end}}(\mathbf{x}) + \hat{p}(z_1, \ldots, z_w), \mathbf{z}) \tag{29}$$

   where at each batch, we randomly choose a different $w$ and $g_{w+1,\ldots,24}(\hat{\mathbf{z}}, \mathbf{z})$ denotes the objective function evaluated only on time points starting from $w + 1$ (since there is no need to evaluate on the first $w$ time points). The leading term $1/(24 - w)$ is meant to take the average cost per hour.

4. **Random action**: We now consider a model which randomly chooses $w$ for every datapoint. There are two shceduling decisions: (1) the schedule up to time $w$ and (2) the schedule after time $w$. The initial schedule is determined by the vanilla end-to-end method, then given observations $z_1, \ldots, z_w$ at time $w$, we use the model $p(\mathbf{x}, z_1, \ldots, z_w)$ to decide the rest of the schedule. Let $\hat{\mathbf{z}}$ denote this combined schedule. The cost of the action is then determined by $g(\hat{\mathbf{z}}, \mathbf{z})$.

5. **Single action**: Here we choose a single fixed across $w$ across all datapoints. We choose this action to be the one that results in the lowest cost on training data. We determine the cost of an action $w$ in the same way as for the *random action* method above.

6. **Optimal action**: For every datapoint, we compute the cost of every possible action $w = 0, \ldots, 24$ and choose the best one.

7. **Endogenous end-to-end**: We now train a model $f^e(\mathbf{x}, w)$ with the goal that using this as a point forecast when making decision $w$. Here, the superscript $e$ denotes endogenous, to separate from previous functions like $f_{\text{end-to-end}}, f_{\text{2-stage}}$. This is done as follows.

   i. We make a point forecast $f^e(w, \mathbf{x})$.

   ii. Given point forecast, we predict $p(\mathbf{x}, f_1^e(\mathbf{x}, w), \ldots, f_w^e(\mathbf{x}, w))$ for time points after time $w$.

   iii. For ground truth, we would observe $z_1, \ldots, z_w$. We would use this instead of $f_1^e(\mathbf{x}, w), \ldots, f_w^e(\mathbf{x}, w)$ when making a forecast for the second stage. That is, we would predict $p(\mathbf{x}, z_1, \ldots, z_w)$ instead.

   iv. The second-stage schedule is given by $p(\mathbf{x}, f_1^e(\mathbf{x}, w), \ldots, f_w^e(\mathbf{x}, w))$ while the schedule given ground truth observations is given by $p(\mathbf{x}, z_1, \ldots, z_w)$. The loss function is then the squared difference between the *cost* of these decisions. Specifically, this is

   $$\left( g_{w+1,\ldots,24}(p(\mathbf{x}, f_1^e(\mathbf{x}, w), \ldots, f_w^e(\mathbf{x}, w)), f^e(\mathbf{x}, w)) - g(p(\mathbf{x}, z_1, \ldots, z_w), \mathbf{z}) \right)^2 \tag{30}$$

   where again $g_{w+1,\ldots,24}(\hat{\mathbf{z}}, \mathbf{z})$ only evaluates the loss starting at time point $w + 1$, omitting the first $w$.

   v. Again, at every batch, we randomly choose some fixed $w$ for all datapoints in the batch.

   Now, given learned $f$ and $p$, we must make a decision according to equation 18. When applied to the electricity scheduling problem, we solve

   $$\min_{w \in \{0, \ldots, 24\}} g_{1,\ldots,w}(f_{\text{end-to-end}}(\mathbf{x}), f^e(w, \mathbf{x})) + $$
   $$g_{w+1,\ldots,24}(p(w, f_1^e(w, \mathbf{x}), \ldots, f_w^e(w, \mathbf{x})), f^e(w, \mathbf{x})) \tag{31}$$

   where the first term evaluates predicted cost of the initial schedule before time $w$ and the second term evaluates predicted cost on the schedule after time $w$.

8. **Cost learner**: Here the model learns the cost of each action $w$ given $\mathbf{x}$. That is, a model $f_{\text{cost}}(w)$ wich predicts

   $$f_{\text{cost}}(\mathbf{x}, w) \approx g_{1,\ldots,w}(f_{\text{end-to-end}}(\mathbf{x}), \mathbf{z}) + g_{w+1,\ldots,24}(p(w, z_1, \ldots, z_w), \mathbf{z}) \tag{32}$$

   the cost of using the vanilla end-to-end schedule up to time $w$, and then the $p$ schedule after time $w$. To make decisions, we choose $w$ which minimizes $f_{\text{cost}}(\mathbf{x}, w)$.

9. **Evaluate on test data:** Finally, for any decision $w$, we evaluate as follows on test data. We use the base vanilla end-to-end schedule for the first $w$ time points. We then observe the ground truth $z_1, \ldots, z_w$ and make predictions $p(\mathbf{x}, z_1, \ldots, z_w)$ to make the rest of the schedule. Formally, the cost is

   $$g_{1,\ldots,w}(f_{\text{end-to-end}}(\mathbf{x}), \mathbf{z}) + g_{w+1,\ldots,24}(p(w, z_1, \ldots, z_w), \mathbf{z}), \tag{33}$$

   exactly the same as the cost-learner's target above.

The full code is available in the supplementary files.

