# OpenReview forum: "End-to-End Learning under Endogenous Uncertainty"
_ICLR.cc/2025/Conference — ICLR 2025 Conference Withdrawn Submission_

### Official Review · Reviewer_tP9r · 2024-11-03

**Soundness:** 1
**Presentation:** 2
**Contribution:** 1
**Rating:** 3
**Confidence:** 5

**Summary:**

The paper proposes a new approach to solve contextual stochastic optimization problems under decision-dependent uncertainty. Their approach focuses on estimating the reward function via a plug-in approach, i.e., replacing the unknown stochastic random variable with a prediction conditional on the context and decision. Since learning the reward function is a non-convex problem, they provide a MIP formulation to solve it exactly for certain classes of plug-ins. The paper also discusses a new class of two-stage stochastic optimization problems related to information-gathering. Finally, they provide some numerics comparing their method to other approaches.

**Strengths:**

The paper's main strength is that they propose an interesting approach to estimating the reward function for offline policy evaluation that seems loosely justified theoretically by Proposition 3.1. They then propose a MIP formulation that provides a tractable way to solve their non-convex learning problem to optimality. It's overall interesting because it directly leverages structure of the stochastic optimization problem compared to more generic offline policy evaluation approaches. This reduces the complexity of learning the reward function which maybe beneficial in small data settings.

**Weaknesses:**

I think the paper can be improved in many different ways. Below outlines some issues that should be addressed:

1. First, **I would not classify their approach as "end-to-end"**, especially in the context of the works [1][2][3]. These works look to directly optimize the decision-loss whereas this paper is primarily focused on learning the reward function. To clarify, an end-to-end approach would look to solve $\hat{f} \in \arg\max_{f \in \mathcal{F}} \mathbb{E}_{\mathbf{z}\sim Z(\hat{\mathbf{w}}_f (\mathbf{x}),\mathbf{x})}\left[ g(\hat{\mathbf{w}}_f(\mathbf{x}), \mathbf{z}) \right]$ where $\hat{\mathbf{w}}_f (\mathbf{x})$ is equation (1) in the paper but parameterized by
$f$. In contrast, **this paper lies more in the offline policy learning for contextual bandits literature** as it proposes the following estimate for the reward function $\hat{r}(w,x) = g(w, \hat{f}\_{\text{end-to-end}} (w,x))$. The policy $\hat{\mathbf{w}} (\mathbf{x})$ is just the policy that optimizes the reward function estimator.

2. The paper claims "We provide theoretical analysis showing that (1) optimally minimizing our proposed objective produces optimal decisions". However, this does not seem to appear anywhere in the paper. The only theory is in section 3, which provides some limited intuition why using plug-in estimator is reasonable (proposition 3.1) and some generic generalization bounds (theorem 3.2). In general Theorem 3.2 seems somewhat problematic. **First**, the authors do not translate theorem into any sort of regret bound to show you obtain optimal decisions. **Second**, the theorem seems potentially incorrect/imprecise since you do not take an expectation of $\mathbf{x}$ so $l(f)$ is still random and not even well defined. My guess is you just want an expectation over all the random variables. **Third**, the assumption that $g(\mathbf{w},\mathbf{z})$ (typo in the theorem) is bounded seems hard to justify if $g$ is Lipschitz and $\mathbf{z}$ is unbounded given the examples discussed in the paper.

3. The information gathering application feels somewhat out of place in this paper. I understand the authors are trying to motivate endogeneity for contextual stochastic optimization, but the application is confusing. Specifically, I do not understand why if you already have data about $Z_w(\mathbf{x})$ why you would need to make a specific decision $w$ to utilize the data in the optimization problem. For the opposite case, if you don't have any data related to $Z_w(\mathbf{x})$ for a specific decision $w$, then how are you supposed to model it in your two-stage contextual stochastic optimization problem. Everything is offline, so I don't understand the information gathering aspect of this example.

4. **The numerics seem extremely limited**. Since the paper doesn't provide any theoretical guarantees in regards to learning "optimal decisions", it would be helpful to have more extensive experimental results and more details about the implementation.  For the assortment optimization experiment, the following is unclear:
- How many trials did you do? It seems like you only did a single sample run, i.e., you only saw the performance of the different methods for a single training set. Training sets are also noisy so it would be nice to see the performance over many training sets and have confidence bands on the average performance.
- It seems like your setting is almost well specified, as in the data generation procedure is contained in the class of linear models you are learning $f$ from. What happens if $z_k$ is not generated from a linear model, but something more non-convex or even a non-continuous function?
- More details and methods would help highlight the benefits of the paper's proposed approach. Why do you not consider an approach where you learn the map $x_n, w_n \rightarrow g(w_n, z_n)$ via a neural network? Is this what reward learning is? Additionally, how do you tune your KNN method?
- What are the run times of your method? Since it's a MIP, it would be helpful to understand the trade-off of the increased computation time vs. improvement in decision quality. Additionally, the paper seems to suggest ways to speed up the computation, so it would be nice to see how these approaches improve the computation time.


___
[1] Adam N Elmachtoub and Paul Grigas. Smart “predict, then optimize”. Management Science, 68(1): 9–26, 2022.
[2] Akshay Agrawal, Brandon Amos, Shane Barratt, Stephen Boyd, Steven Diamond, and J Zico Kolter. Differentiable convex optimization layers. Advances in neural information processing systems, 32, 2019.
[3] Brandon Amos and J Zico Kolter. Optnet: Differentiable optimization as a layer in neural networks. In International conference on machine learning, pp. 136–145. PMLR, 2017.

**Questions:**

Below are questions that perhaps don't fall directly in the categories of the weakness section:

1. Is $\mathbf{x}$ fixed or random? It doesn't seem well defined in the paper which can interfere with the theory in the paper.
2. Equation (7) is confusing. Is $\bar{\mathbf{z}}$ random on the left hand side like it is on the right hand side? If so, then I do not see how the $\approx$ makes sense since if $\bar{\mathbf{z}}]$ varies a lot then you shouldn't ever be able to estimate the constant on the right hand side.

---

> ### Author Response · Authors · 2024-11-24
> **Author Response (part 1)**
>
> Thank you very much for the thoughtful comments, reviews and time carefully reading the paper. These have greatly improved the paper. We would like to address each of your points, and have also revised the paper based on your suggestions.
>
> 1. We agree this is not strictly end-to-end in the same way as existing work like those you cite. We also considered the formulation you described when we first began work --- this is definitely the most direct connection to current end-to-end work. Unfortunately, it is not solvable as it is. Specifically because we cannot evaluate the expectation as the distribution
> $Z(\hat{w}(f(x)), x)$
> is unknown and we have no data for it. This approach would require knowing the counterfactual distribution for a different action taken. In exogenous case, this distribution was only $Z(x)$ and hence historical observations sufficed but here it depends on actions $w$.
>
> While our paper does have strong connections to offline policy learning, our approach still takes ideas from the traditional end-to-end methods. Specifically, that ${\hat{f}_{end-to-end}}$
>
> aims to make the best prediction (of the random variable $z$) that results in the best approximation of the reward. In the traditional end-to-end setting, $\hat{f}_{{end-to-end}}$ aims to make the best prediction that results in the best decision. This was the connection we were trying to express.
>
> 2) Our claim that "optimally minimizing our proposed objective produces optimal decisions"is perhaps too generic and we have re-framed this in the revised paper. However, optimally solving for $f^*$ to achieve equation (2) indeed produces optimal decisions. This is primarily what we referred to.
>
> Overall, we restructured section 3 and placed the results in the previous section to better convey the meaning. First, we show that $f^*$ as described in equation (2) can exist in the first place. This is given by proposition 3.1 (now proposition 2.1 in the revised draft). Second, the hypothesis class likely does not contain $f^*$ but we can increase its complexity to get better results and be closer to $f^*$. Theorem 3.2 (now 2.3) explains the tradeoff to increasing hypothesis class complexity when we have limited data. We hope the revised paper provides this clearer message.
>
> We agree our paper does not make any theoretical guarantees in regards to learning "optimal decisions", although this would be an interesting future direction. We have included many new experiments to at least empirically show strong performance.
>
> Thank you for bringing up some of the smaller issues in the theorem. Indeed, the expectation should be over $x$ as well. Second, you make a good point, we do implicitly assume that $z$ is bounded (even in the examples we give, it would be unnatural to have unbounded demand for instance). We have amended the paper with these points.

---

> > ### Author Response · Authors · 2024-11-24
> > **Author Response (part 2)**
> >
> > 3) It might be easiest to understand the information-gathering setting through the problem in the computational experiment.
> >
> > In the information gathering setting, $z$ is independent of the actions taken (in the numerical experiments, $z$ is electricity of demand for each of the next 24 hours). Before taking an action, $z$ is unknown. However, we can query to gain information about $z$. In the experiment, this amounts to waiting $t$ hours and observing the first $t$ entries of $z$. The first decision (the amount of time $t$ to wait) can be solved by our approach in the endogenous setting. Intuitively, this is because the observations we make on $z$ depends on our decision $t$. Similarly in the endogenous setting, the distribution of $z$ depends on our action.
> >
> > In general, the information gathering setting assumes we can make a first-stage decision/query to gain information about some entries of $z$ before we make the operational second-stage decision. Information-gathering specifically refers to this two stage nature of the decisions (first querying information about $z$, then making an operational decision given this information). We have done our best to make this clearer in the revised paper as well.
> >
> > 4) We have added additional computation results to address your questions. We have taken your advice on running more experiments across multiple datasets, and across a wider range of parameters. We also consider the mis-specified setting. And yes, we do consider learning a map $x_n, w_n \to g(w_n, z_n)$, and yes this is indeed the reward-learning benchmark. We have done our best to clarify the benchmarks we used in the revised paper. For the KNN method, we chose $K$ by trying a variety of values ($2, 5, 10, 20, 50$) and choosing the one that best approximates cost and demand.
> >
> > Furthermore, after running fairly extensive experiments, solving equation (3) simply by gradient descent (in spite of non-convexity) performs very well, and very simple to implement compared to the other ideas in section 2.1 (the MIP and sampling method). For better clarity and continuity of the paper (as well as to save space for more computational results) we decided to move section 2.1 to the appendix.
> >
> > We wrote a more detailed description of these new computational results in a general response, since it would be of interest to all reviewers as well. Please see the revised draft of the paper as well for details.
> >
> >
> > To address your final question about equation (7), thank you for catching this. There is a typo, and the expectation around $\bar{z}$ is missing on the left hand side.
> >
> > We would like to thank you again for your feedback.

---

> > > ### Comment · Reviewer_tP9r · 2024-11-24
> > >
> > > Thank you for your response. I appreciate the updates!
> > >
> > > I understand your explanation of why you positioned your work as "end-to-end." However, I still think it feels deceptive since you are not making end-to-end learning decisions but instead predictions of the function $f$ to improve **reward prediction accuracy**. There are still no theoretical guarantees that this method produces better decisions.
> > >
> > > That being said, I think you most likely need to show that your results are robust numerically. The current set of experiments still feels very limited and not very robust. Below are some issues you should address:
> > >
> > > 1. **Strength of benchmarks:** Most of your benchmarks do not tune any hyper-parameters (for KNN you only search over 5 choices of a single hyper-parameter). Arguably, your method is lower complexity, so it feels unfair to artificially increase the complexity of the benchmarks so that your lower complexity method performs better in high noise settings. My original guess was that i) the percent improvement reported in Table 3 could be due to the fact that your higher complexity methods overfit to the noise and end up doing worse, and ii) your lower complexity method doesn't change in performance. More strangely, when I checked your appendix, your method does better when there is more noise, which seems counter-intuitive. To improve the robustness, I would test your method over more than five datasets (and plot some confidence intervals) and enhance the strength of the benchmarks.
> > > 2. **Choice of Hypothesis Class:** As you mentioned in your response, $f^*$ is unlikely to be contained in the hypothesis class you are searching over. How do you choose the "right" hypothesis class, and does increasing the complexity of the hypothesis class eliminate the benefits your method seems to provide?
> > > 3. **Computational Complexity Comparisons:** Do you use gradient descent or solve your MIP in your experiments? Does obtaining the optimal solution of the MIP improve the performance of your method? What's the trade-off between performance and computational complexity? Answering these questions would better help justify the MIP formulation you provided. It might also justify removing the formulation entirely.
> > > 4. **Sampling Robustness:** The major challenge in learning the reward function under endogenous settings is that the sampling is biased. Your current theory assumes your samples are generated I.I.D. from some distribution. How does your method perform when your data is not generated from an I.I.D. setting? Does it perform better or worse? I am worried that because the true $f^*$ does not represent the expected demand structure, you may interpolate poorly and do worse than other benchmarks that may try to address the sampling bias.
> > > 5. **Strange results:** Coming back to point 1, the results in the appendix, Figure 3b in particular, seem to show that when you increase the amount of data, all benchmark methods perform worse. That seems counter-intuitive since, the prediction accuracy of these methods should be getting better.  Is there some issue with the experiments?
> > >
> > > Finally, Section 3 still feels like it distracts from the overall goal of the paper since there needs to be a lot of discussion for the modeling choices, specifically for why you chose the form you did for estimating the reward function. For example, why do you need to predict the samples you will obtain from waiting in your electricity scheduling problem. Why can't you just learn p(x,w)? Maybe the answer is obvious, but it seems to be an orthogonal concern relative to the rest of the paper.

---

> ### Author Response · Authors · 2024-11-25
> **Response**
>
> Thank you for your quick response and very helpful feedback. We have amended the computational results with your comments.  To address each of your points:
>
> points 1/2) Specifically, we run each experiment on 20 datasets, and report percent error compared to the optimal solution as well as standard deviations.
>
> We would like to note that we use the same model architecture for each approach. So, our method does not use a simpler model. Using the same complexity model, we find our approach performs better. Your comments also gave us the idea of comparing method performance as we increase the complexity of the predictive model --- in this case the width of the hidden layer of the network. We see that the reward-learning methods requires 500-1000 hidden layer width to achieve the same accuracy as our approach which only needs width 30.
>
> For the noise experiments, there was likely some mistake in the implementation, and we have fixed this. Indeed, as noise increases performance becomes worse for all methods. We still observe our method performs better than the remaining methods at higher noise values. For the reward learning method we compared against both a network of width 50 (the same we use for our approach) and one of width 500. Even with width 500, the reward-learning method under-performs.
>
> point 3) We decided not to include the MIP formulation since it is quite expensive to compute, and from the experiments we find simply using gradient descent instead nearly optimally solves the problem. We moved this to the appendix in case it is of interest.
>
> point 4) We agree that our method may not be best-suited when there is sampling bias. This may be remedied by existing work in the literature, which for instance constrains the decision-space to be "in-distribution" with the training data. This would maintain the accuracy of our method on the constrained space of decisions. This is similar in idea to offline reinforcement learning work which constrains the distance between state-action visitation of the policy and the state-action pairs contained in the data. See for instance [1]. This is perhaps out of scope of our paper, although we are happy to add discussion on this point in the paper.
>
> point 5) The new results don't exhibit any strange behavior. As data increases, all methods perform better overall; as noise increases, all methods perform worse.
>
> We believe section 3 is an interesting application -- for instance, it combines with the traditional exogenous end-to-end method as the endogenous aspect appears due to the two stage nature of the problem. This type of problem has not been studied in the literature as far as we know. We could potentially move this section to the appendix if other reviewers feel it is distracting. To answer your question, yes one can simply learn the value/reward/cost of making a waiting decision (this is the reward-learning approach we compare against). Similar to the setting considered in the previous part of the paper, there is value in making an intermediate prediction of what the observed samples would be.
>
> [1] Off-Policy Deep Reinforcement Learning without Exploration. Scott Fujimoto, David Meger, Doina Precup

---

### Official Review · Reviewer_5dkt · 2024-11-04

**Soundness:** 2
**Presentation:** 2
**Contribution:** 2
**Rating:** 5
**Confidence:** 3

**Summary:**

The paper provides a formulation in "predict then optimize" situations where the optimization decisions affect the random realizations. The paper solves a version of the above problem for some structured cases via MIP optimization.

**Strengths:**

The prediction and then optimization problem is relevant to operations research and other areas. The mixed integer programming formulation of the above problem is novel. The paper also has reasonable empirical work to support their theory.
The introduction and motivation are well written and play a good role in building anticipation for the further sections.

**Weaknesses:**

1)The introduction is well written, but it motivates a very general setting of the problem, while the paper seems to address only a very structured version of the general problem (with examples in lines at the beginning of section 2).

2) The setting as motivated is similar to "Performative Prediction" https://proceedings.mlr.press/v119/perdomo20a.html. The authors should consider contrasting their work with this and other works in performative machine learning.

3) Realizability assumptions are not explicitly made (e.g. f* should be part of the function class for (3) as a necessary condition). Other assumptions may be required based on the exact nature of the "sampling method"

4)Why is the non-convexity of z is an issue (Line 183) in solving (3). It is not clear the analogy made, r_n is not defined.

5) The integer program appears all of a sudden; it is not well motivated, and all the variables are not fully defined in the main text. Why is this the chosen model for the given applications ?

6)The sampling method is fully described, e.g., does perturbing the mean squared predictor help? Section 2.1 seems to be overall hurriedly written or hurriedly shortened just to fit the page limit

7) I do not fully follow the message that section 3 is trying to convey. While for the applicability of proposition 3.1, one needs a function of having "infinite expressivity" (suitability defined), how does the Rademacher complexity of such a function class behave? So, in some sense, the bounds of Theorem 3.2 only make sense when $N = \infty$ (if we can suitably show the R_N(F) falls down suitably fast)

8)In the information gathering section, the setting is not well described, as it seems that the realization of z is fixed (but unknown and something that can be queried). Can you clarify

**Questions:**

Please refer above.

---

> ### Author Response · Authors · 2024-11-24
> **Author Response**
>
> We appreciate your time in carefully reading the paper and your thoughtful comments. These have been helpful in restructuring parts of the paper to make it clearer and easier to read. In addition, we would like to let you know of some new computational results we ran as suggested by other reviewers. We hope you find these interesting as well. We discuss this in the common response as well as in the revised draft.
>
> To address each of your points:
>
> 1) Our approach is indeed focused on more structured decision-making problems where objectives, constraints etc. are well-defined. We've made this more clear from the beginning of the introduction to better place the setting of the paper. All of the examples we give from the start of the introduction fall within the setting we consider in the rest of the paper.
>
> 2) This is an interesting area of work, thank you for bringing it to our attention. We have included a short discussion on this in the related works section. In this area of work, a performative prediction is one which affects the outcome/distribution of the target. For example, this area would focus on problems like how traffic predictions will affect driver behavior and hence affect traffic itself.
> This is similar to our setting where instead the decision we take affects the distribution of the target. Another difference in performative learning is the online aspect whereas we consider the offline setting.
>
>
> 3) Yes, for the optimal solution to the optimization problem in (3) to be the wanted $f^*$ which satisfies (2), we do need to make the assumption that $f^*$ is contained in our hypothesis class. Proposition 3.1 (now proposition 2.1 in the revised draft) shows that $f^*$ exists in the first place, although it may fall outside the hypothesis class. Nevertheless, solving problem (3) still brings value. In the new experiments we consider the mis-specified setting and see strong results. We have added this as a point of discussion in the paper following equation (3).
>
> 4,5,6) The non-convexity of the objective function is an issue from the point of view of solving the problem to optimality. The integer program is mean to address the issue. However, we agree section 2.1 feels somewhat out of place. After running fairly extensive experiments, solving equation (3) simply by gradient descent (in spite of non-convexity) performs very well, and very simple to implement compared to the other ideas in section 2.1. For better clarity and continuity of the paper (as well as to save space for more computational results) we decided to move section 2.1 to the appendix.
>
> 7) Thank you for pointing this out, section 3 indeed does not flow particularly well. We have moved each proposition/theorem to a more appropriate place within the paper to connect better. In particular, we describe the following flow.
>
> First, that $f^*$ as described in equation (2) can exist in the first place. This is given by proposition 3.1 (now proposition 2.1 in the revised draft). Second, the hypothesis class likely does not contain $f^*$ but we can increase its complexity to get better results and be closer to $f^*$. Theorem 3.2 (now 2.3) explains the tradeoff to increasing hypothesis class complexity when we have limited data. We hope the revised paper provides this clearer message.
>
> 8) In the information gathering setting we consider a slightly different problem than the initial endegenous setting. But we show that it can reduce to the endogenous setting. In the information gathering setting, $z$ is independent of the actions taken (in the numerical experiments, $z$ is electricity of demand for each of the next 24 hours). However, we can query to gain information about $z$. In the experiment, this amounts to waiting $t$ hours and observing the first $t$ entries of $z$. The first decision (the amount of time $t$ to wait) can be solved by our approach in the endogenous setting. Intuitively, this is because the observations we make on $z$ depends on our decision $t$. Similarly in the endogenous setting, the distribution of $z$ depends on our action.
>
> In general, the information gathering setting assumes we can query to gain information about some entries of $z$.

---

> > ### Comment · Reviewer_5dkt · 2024-11-28
> > **Thanks for your rebuttal**
> >
> > A lot of my queries were answered, I have raised my score

---

### Official Review · Reviewer_6pyf · 2024-11-04

**Soundness:** 3
**Presentation:** 3
**Contribution:** 3
**Rating:** 5
**Confidence:** 3

**Summary:**

The authors study an end-to-end approach that tackles endogenous problems, specifically contextual stochastic optimization problems under decision-dependent uncertainty. Their approach has a first information gathering stage and a second decision stage. They provide generalization bounds between in-sample and out-of-sample cost, and they empirically demonstrate the advantages of their approach.

**Strengths:**

For their proposed method, the authors provide both theoretical analysis and empirical results on synthetic and real-world data.

The paper is well-organized and easy to follow.

**Weaknesses:**

The authors only compared against one end-to-end baseline, while there has been many advances in this area [1]. [1] is a python library that  includes implementation of several end-to-end learning algorithms and testing environments. [2] benchmarks a few end-to-end learning methods. [3] and [4] are some examples of existing end-to-end methods that I am sure the authors are familiar with.

Although not all existing end-to-end methods may apply to the experimental settings the authors considered, I encourage the authors to compare against at least one or two more end-to-end algorithms and especially the recent state-of-the-art, to showcase the benefits of proposed method in handling endogeneity.

Apart from baselines, the experimental settings are limited. For example, table 1 only shows results in one single setting. This is synthetic data experiment, and thus it should be straight forward to obtain results on a wide range of settings (e.g. different number of products, costs, and noise levels). Having results on at least several different settings would greatly improve the work.

The evaluation metrics currently do not include standard errors or confidence intervals. Thus, it is not clear whether there is statistically significant difference between the proposed method and the baselines. Providing either paired t-test results or standard errors would help showcase the advantages of the proposed method.

[1] Tang, B. and Khalil, E.B., 2024. Pyepo: A pytorch-based end-to-end predict-then-optimize library for linear and integer programming. Mathematical Programming Computation, 16(3), pp.297-335.
[2] Geng, H., Ruan, H., Wang, R., Li, Y., Wang, Y., Chen, L. and Yan, J., 2023. Rethinking and Benchmarking Predict-then-Optimize Paradigm for Combinatorial Optimization Problems. arXiv preprint arXiv:2311.07633.
[3] Amos, B. and Kolter, J.Z., 2017, July. Optnet: Differentiable optimization as a layer in neural networks. In International conference on machine learning (pp. 136-145). PMLR.
[4] Kallus, N. and Mao, X., 2023. Stochastic optimization forests. Management Science, 69(4), pp.1975-1994.

**Questions:**

What are the technical difficulties and novelties in the proof of theoretical results? Specifically, what are the challenges in applying standard proof techniques using Rademacher complexity to the problem setting considered in this paper? It would be nice to highlight in section 3.

---

> ### Author Response · Authors · 2024-11-24
> **Author Response**
>
> Thank you for your time in reading the paper and the constructive feedback. It has been very helpful in improving the paper. We have taken your advice on running more experiments across multiple datasets, and across a wider range of parameters. We wrote a more detailed description of these new results in a general response, since it would be of interest to all reviewers as well. Please see the revised draft of the paper as well for details.
>
> In terms of benchmarking against other end-to-end approaches as you mentioned, this is not possible since they are not applicable to our endogenous setting. We have a discussion of why this is the case in section 2.2. Intuitively, every existing end-to-end approach, including all that you cite, one needs to train by being able to evaluate the cost of any decision for a given training datapoint/observation (exogenous demand setting). However, in the endogenous uncertainty setting we consider in this paper this is not possible because the distribution of the random variable changes for every decision. This is in large part the reason we studied this endogenous problem and proposed our method.
>
> We did add another benchmark and  compared against a guassian process method which learns to predict the cost of a decision directly (skipping the intermediate prediction of the random variable).

---

### Official Review · Reviewer_CBgf · 2024-11-13

**Soundness:** 3
**Presentation:** 2
**Contribution:** 3
**Rating:** 3
**Confidence:** 3

**Summary:**

The manuscript proposes a joint prediction-and-optimization framework for the contextual stochastic optimization problem under endogenous uncertainty, and for a class of two-stage information-gathering problem. The framework has been tested using two specific examples and the results show the newly proposed framework outperforms baselines.

**Strengths:**

The manuscript provides a novel method to estimate the reward function when decision-dependent uncertainty is present. It also provides theoretical proof to the achievability of the formulations, even though loosely. Empirical experiments are also conducted to further prove the performance of the algorithm.

**Weaknesses:**

1. More experiments are needed to demonstrate the performance of the method.
2. The experimental results are not persuasive enough by simply comparing with the shown baseline models. It would be necessary to include other advanced methods.

**Questions:**

See above.

---

> ### Author Response · Authors · 2024-11-24
> **Response**
>
> Thank you for your time in reading our paper. We agree that further experimental results are crucial to demonstrate the performance of the method. We have done our best in the short review period to add an extensive set of new results. Please see our general response as well as the revised draft of the paper for details.
>
> If there are any other specific methods you would like us to compare against which can be applied to our problem setting, please let us know.  With the new computational results, we consider 5 benchmarks. (1) a \emph{predict-then-optimize} method, also known as a \emph{two-stage} method, which
> train a model to learn the uncertainty (demand in this case) as a function of actions (inventory in this case). (2) A \emph{cost-learning} method which trains a model to learning cost directly as a function of $\mathbf{w}$. This does not take into account the intermediate demand data or the structure of the cost function. It only observes the final cost of a decision. (3) Similar to method (2), we predict cost using a \emph{gaussian process} method instead. (4) A K-NN-based method which, for any action, approximates cost by computing the average of the costs of the $K$-nearest neighboring actions. (5) A K-NN method which instead predicts demand. This is a fairly broad range of methods, both considering predicting intermediate demand as a function of actions, learning cost directly as a function of actions, and using both parametric and non-parametric methods for each.

---

### Author Response · Authors · 2024-11-24
**Common Response**

We would like to thank all of the reviewers and chairs for their time in reviewing the paper and for the thoughtful comments. We believe that have greatly improved the paper. We have done our best to address all of your comments. We have also run extensive new computational experiments based on your feedback.

We have revised the paper with these new experiments as well as with the rest of the feedback from everyone. We would like to highlight the computational results in this common response since we believe everyone will be interested in seeing these.

We consider the assortment optimization problem from a variety of more angles. \textbf{Overall, our method outperforms all other methods in high noise, across any  cost configuration, and when given sufficient data.} Improvement range from 10\% to 40\% in high noise.

We evaluate the approaches by constructing 5 datasets by randomly initializing the parameters of the data-generation model (please see section 4 for details). We report two metrics: the average cost of each method across the 5 datasets, as well as the number of datasets that each method performed better than all other methods. We refer to the latter as the number of \emph{wins} each method has. We compute these metrics as vary various parameters: (i) the amount of training data available, (ii) the backorder cost to holding cost ratio, (iii) the amount of noise in the data. For brevity, we report the number of wins of our approach (5) for each set of experiments as well as the average percent improvement of our approach against the next best method on each dataset. Specifically, improvement refers to $(c - c_{{end-to-end}})/c$ where $c$ is the average cost of a competing method across all datasets and $c_{{end-to-end}}$ is the average cost of our proposed approach.


We compare against the following methods (1) a \emph{predict-then-optimize} method, also known as a \emph{two-stage} method, which
train a model to learn the uncertainty (demand in this case) as a function of actions (inventory in this case). (2) A \emph{cost-learning} method which trains a model to learning cost directly as a function of $\mathbf{w}$. This does not take into account the intermediate demand data or the structure of the cost function. It only observes the final cost of a decision. (3) Similar to method (2), we predict cost using a \emph{gaussian process} method instead. (4a) A K-NN-based method which, for any action, approximates cost by computing the average of the costs of the $K$-nearest neighboring actions. (4b) A K-NN method which instead predicts. (5) We use our proposed method solving equation 3 in the paper. Each of the methods (1), (2), (5) use the same model architecture for making predictions.

This also consider the mis-specified setting as we increased the complexity of the data-generation process (please see the experiment section of the paper for details). We use a feedforward neural network with a fully-connected layer of width 100. Each method is trained with the same parameters until convergence.

---

> ### Author Response · Authors · 2024-11-25
> **More Computational Results**
>
> We would like to update the computational results based off of new experiments as suggested from follow-up conversation with reviewer tP9r. Please see the revised paper.
>
> We now run each experiment on 20 datasets, and report percent error compared to the optimal solution as well as standard deviations. Using the same complexity model, we find our approach performs better. We also run a new experiment comparing method performance as we increase the complexity of the predictive model --- in this case the width of the hidden layer of the network. We see that the reward-learning methods requires 500-1000 hidden layer width to achieve the same accuracy as our approach which only needs width 30.
>
> Similarly for the noise experiments, we still observe our method performs better than the remaining methods at higher noise values. For the reward-learning method we used both a network of width 50 (the same we use for our approach) and one of width 500. Even with the width 500 network, the reward-learning method under-performs compared to our approach. We hope these results strengthen the claim that our approach can perform better or as well as other methods while using a lower-complexity model.

---

### Author Response · Authors · 2024-12-04

We would like to thank all of the reviewers again for your comments and thoughtful suggestions. We have done our best to respond and address your concerns and we feel it has been very helpful in improving the manuscript. We have updated it and submitted with major changes in the experiments in section 4 with additional results in appendix C.1 according to your suggestions.

---

### Note · Authors · 2025-02-25

I have read and agree with the venue's withdrawal policy on behalf of myself and my co-authors.

---

### Meta-Review · Area_Chair_7hoA · 2024-12-22

**Metareview:**

The consensus of the reviewers point to a lack of satisfactory theoretical guarantees and limited experiments. The authors have added some new experiment results. However, post rebuttal, there's still a lukewarm sense of the paper still being a work-in-progress. Underlying the negative sentiment of the reviewers is a single result that truly differentiates this work's value. I suggest the authors take this in mind when revising for the next conference.

**Additional Comments On Reviewer Discussion:**

NA

---

### Decision · Program_Chairs · 2025-01-22

Reject